# Exploiting Topology of Protein Language Model Attention Maps for Token Classification

## Abstract

In this paper, we introduce a method to extract topological features from transformer-based protein language models. Our method leverages the persistent homology of attention maps to generate features for token (per amino-acid) classification tasks and demonstrate its relevance in a biological context. We implement our method on transformer-based protein language models using the family of ESM-2 models. Specifically, we demonstrate that minimum spanning trees, derived from attention matrices, encode structurally significant information about proteins. In our experiments, we combine these topological features with standard embeddings from ESM-2. Our method outperforms traditional approaches and other transformer-based methods with a similar number of parameters in several binding site identification tasks and achieves state-of-the-art performance in conservation prediction tasks. Our results highlight the potential of this hybrid approach in advancing the understanding and prediction of protein functions.

## 1 Introduction

Proteins play an essential role in a multitude of biological processes: facilitating chemical reactions, transporting molecules, mediating cellular communication, and providing structural support to both cells and entire organisms. This astonishing functional diversity of proteins is uniquely encoded in their amino acid sequences. With about 20 different standard types of amino acids available, an infinite number of proteins can be generated by varying their sequence arrangements. These amino acid sequences are known as the primary structure of the protein or 1D encoding. Within biological organisms, proteins are spatially coiled, bent, and spontaneously folded due to the interaction of amino acids, resulting in a specific three-dimensional molecular structure known as the tertiary structure of the protein, or 3D structure.

Several unique properties of proteins can be derived from their 3D structure (Kucera et al., 2024; Sun et al., 2024; Wang et al., 2022a; Zhang et al., 2022). However, despite the progress in solving the protein folding problem (Hie et al., 2022; Jumper et al., 2021), existing methods remain computationally demanding. Another approach to predicting protein functions (Kim and Kwon, 2023; Marquet et al., 2022; Rao et al., 2019; Wang et al., 2022b; Xu et al., 2022) involves the use of the protein language models (pLMs) (Ahmed et al., 2020; Heinzinger et al., 2023; Lin et al., 2022; 2023; Rives et al., 2021). These models are based on the transformer architecture (Vaswani et al., 2017) and take into account the sequential nature of proteins. Statistical patterns within evolutionarily related protein sequences provide insights into their structure and function (Altschuh et al., 1988). This connection arises because the properties of a protein impose constraints on the evolution of its sequences. As pLMs meld knowledge of protein sequences with language modeling techniques, their importance has grown significantly. However, the extent to which these models truly grasp the underlying biophysics of protein structure and, by extension, their functions, remains an open question.

Several approaches have been proposed for analyzing the attention maps of models trained on protein sequences (Bhattacharya et al., 2021; Vig et al., 2020). (Bhattacharya et al., 2021) demonstrated that attention mechanisms offer a systematic and principled model of protein interactions, rooted in the inherent properties of protein family data. (Vig et al., 2020) conducted a comprehensive analysis

of the interpretability of protein language models, particularly those based on the BERT architecture. Their study focused on understanding how attention mechanisms within these models capture and represent the intricate features of protein sequences. According to (Vig et al., 2020) findings, the attention maps generated by the models: highlight amino acid pairs distant in sequence but close in structure, as indicated by correlations with pairwise contacts, highlight binding sites within proteins and capture local secondary structure, revealing patterns corresponding to structural motifs like alpha-helices and beta-sheets. This results suggest that protein language models can infer structural proximity from sequence data alone, recognize functionally important sites essential for protein activity, and detect common structural motifs inherent in protein sequences. This demonstrates the capability of attention maps to uncover intricate structural features solely from sequence information.

In this work, we take a step further by proposing a novel approach to analyzing protein language model attention maps using tools from topological data analysis. Specifically, we introduce a minimum spanning tree (*MST*)-based method for *Res*idue classification, called *RES-MST*. We empirically demonstrate that topological features defined on minimum spanning trees derived from attention matrices encode structurally significant information about proteins. Moreover, we extend interpretability research of protein language model attention maps by performing a quantitative analysis of the correspondence between nodes of the minimum spanning trees and residues with high conservation levels.

Topological Data Analysis (Barannikov, 1994; Chazal and Michel, 2017; Zomorodian, 2001) is a field focused on the numerical characterization of multi-scale topological properties of data, including graphs and point clouds. Self-attention maps can naturally be represented as fully connected weighted graphs, where weights, derived from attention scores, indicate similarity (or dissimilarity) in some sense between nodes (amino acids). Previously, approaches to compute topological features from attention matrices of foundation models were explored in the fields of NLP and speech processing (Cherniavskii et al., 2022; Kushnareva et al., 2021; Tulchinskii et al., 2022), focusing solely on sequence classification. In contrast, our work aims to make predictions at the individual token (amino acid *residue*) level, enabling us to address crucial per-residue tasks. Common examples of per-residue tasks include: identifying which residues are likely to be involved in binding with other molecules such as ligands, DNA, or other proteins (binding sites) and predicting the level of evolutionary changes at specific residue positions (conservation) which might affect the protein's structure and function. Understanding the characteristics of each residue can reveal detailed mechanisms of protein function, interaction, and stability that are not apparent at the whole-protein level. For example, knowing which residues are involved in binding sites or are critical for structural integrity allows for more precise targeting in drug design (Lu et al., 2022; Pei et al., 2024). This specificity can lead to the development of drugs with higher efficacy and fewer side effects. Additionally, diseases caused by genetic mutations often result from changes in single amino acid residues that affect protein function. Per-residue binding and conservation predictions can help identify which amino acids are likely to be exposed to that.

In summary, we make the following contributions:

- We introduce a novel non-parametric framework specifically designed to convert attention matrices from transformer models into topological features tailored to token-wise classification. To the best of our knowledge, this is the first time that Topological Data Analysis has been directly applied to classification on a per-token basis;

- Our work is the first one to study the topology of attention maps in protein language models, as opposed to applying it directly to 3D structures (Dey and Mandal, 2018; Koseki et al., 2023; Swenson et al., 2020);

- We perform a quantitative analysis to demonstrate that the topological structures derived from attention maps, particularly, the minimum spanning tree, capture structurally significant information about proteins;

- Finally, we find out that our topological features are informative and, being combined with traditional pLM's vector representations, outperform sequence-based state-of-the-art methods on several tasks such as prediction of binding sites and conservation.

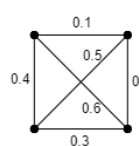

Figure 1: A fully connected weighted graph.

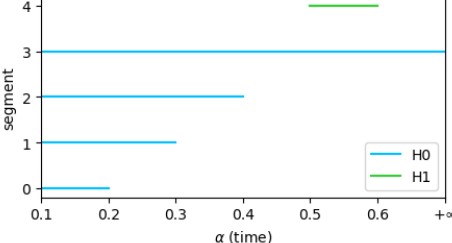

Figure 2: A barcode of the weighted graph from Figure 1. Four $H_0$ bars correspond to 4 nodes (connected components for $\alpha = 0$); one $H_1$ bar corresponds to a cycle.

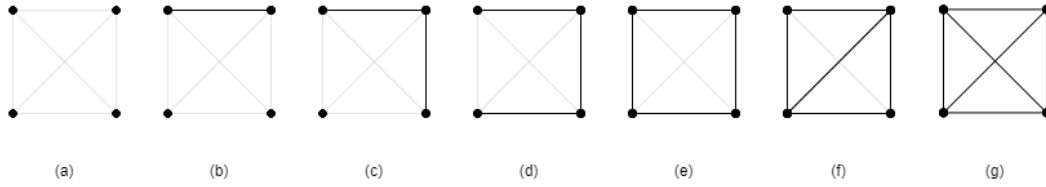

Figure 3: A filtration of a simplicial complex of the graph from Figure 1. (a)-(f): its subgraphs when ranging $0 \leq \alpha \leq 0.6$. (d): a minimum spanning tree of the graph, $\alpha = 0.3$. (e)-(f): for $\alpha = 0.4$ a cycle appears, then it disappears for $\alpha = 0.5$.

## 2 BACKGROUND ON TOPOLOGICAL DATA ANALYSIS

Topology is often considered to describe the "shape of data". In this Section, we give a high-level introduction to the flagship tool of Topological Data Analysis - a persistent homology. For a detailed explanation, we refer a reader to (Dey and Wang, 2022).

**Simplicial homology.** The central object of this work is a self-attention matrix. After some symmetrization, we treat it as a fully connected weighted undirected graph $\mathcal{G}$ having tokens as vertices. For a weighted graph $\mathcal{G}$, we are interested in studying its subgraphs where only edges with associated weights $w_{ij} \leq \alpha$ are taken, see Figure 1 and Figure 3.

The subgraphs $\mathcal{G}^\alpha$ might have distinct topology, that is, number of connected components, cycles, voids, etc. These features are precisely characterized by $k$-th *Betti numbers* $\beta_k \in \mathbb{N}_0$ by means of a simplicial homology. Formally, the Vietoris-Rips simplicial complex[1] is defined as:

$$\text{VR}_\alpha(\mathcal{G}) = \{\{i_0, \ldots, i_k\}, i_m \in \text{Vert}(\mathcal{G}) \mid w_{i,j} \leq \alpha\},$$

where $\text{Vert}(\mathcal{G})$ - is the vertex set of the graph $\mathcal{G}$. The simplicial complex can be considered as a generalization of a graph having high-order relations of vertices. Betti number $\beta_k$ is a dimensionality of a homology group $H_k$ of $\text{VR}_\alpha(\mathcal{G})$.

**Persistent homology.** A natural issue is the necessity of choosing a value for $\alpha$. It is solved by a persistent homology, which permits to analyse $\alpha \in [\alpha_{min}, \alpha_{max}]$ all together. By considering $\alpha_1 \leq \alpha_2 \leq \ldots \leq \alpha_m$, we get a nested sequence $\text{VR}_{\alpha_1}(\mathcal{G}) \subseteq \text{VR}_{\alpha_2}(\mathcal{G}) \subseteq \ldots \subseteq \text{VR}_{\alpha_m}(\mathcal{G})$, which is called a *filtration*.

The values of $\alpha$ when topological features (like connected components and a cycle on Figure 3) appear and disappear can be paired and form a segment $(\alpha_m, \alpha_n)$. A multiset of such segments is called a *persistence barcode* and it describes topological features at multiple scales. The longer segment is, the more "persistent" (distinguishable from noise) is the corresponding topological feature, see Figure 2 for an illustration. The whole theory is dubbed a *persistent homology* (Barannikov, 1994; Chazal and Michel, 2017; Zomorodian, 2001).

---

[1]Simplicial complex is a family of sets that is closed under taking subsets.

Computation of a persistence barcode it computationally demanding. Generally, the barcode computation is at worst cubic in the number of simplexes involved. In practice, the computation is faster since the boundary matrix is typically sparse for real datasets. Ablation studies comparing the $H_0$ and $H_1$ persistence homology topological features are provided in the Appendix section B.2.

We derive our method RES-MST based on the $H_0$ persistence homology. The algorithm for building an $H_0$ persistence barcode is essentially an algorithm for finding a Minimum Spanning Tree (MST) of a weighted graph. A bar in the $H_0$ barcode corresponds to an edge in MST, because both are constructed by incrementally connecting components in a graph based on ascending edge weights. In $H_0$ persistent homology, an interval represents the lifespan of a connected component, ending when it merges with another, which directly maps to the addition of an edge in the MST that connects two disjoint components. This equivalence arises because both processes prioritize edges by weight to form a single connected structure. See Section 3.5.3 from Dey and Wang (2022) for more details.

The basic statistics computed over the edges in the MST, such as the minimum, maximum, sum, and mean of edge weights, align with those derived from the $H_0$ barcode because the intervals in the barcode encode the same edge weights. The length of each interval in the $H_0$ barcode corresponds to the weight of an edge in the MST. Therefore, summarizing these weights through statistics directly captures the key features of the $H_0$ barcode, making the two representations equivalent in terms of the structural information they encode. If one is interested only in $H_0$ barcodes, one can use Kruskal's algorithm with a complexity $O(E \log(E))$, where $E$ is a number of graph's edges.

## 3 A TOPOLOGY-BASED FRAMEWORK FOR ATTENTION MATRICES OF PLMS

Protein language models (pLMs) pretrained on extensive protein sequence corpora have demonstrated impressive results in predicting protein function and structure. These models, typically based on transformer architectures, are trained using a masked protein modeling task. In this task, partial residues are masked in the input sequence and are predicted based on their context. One such state-of-the-art protein language model, ESM-2 (Lin et al., 2022), is trained on protein sequences from the UniRef database (Suzek et al., 2015). In this model, given an input protein sequence, 15% of amino acids are masked, and ESM-2 is tasked with predicting these missing positions (Devlin et al., 2018). Although the primary training objective involves predicting these missing amino acids, achieving high accuracy necessitates that the model learn complex internal representations of its input. These representations learn secondary structure prediction, binding site prediction, and contact prediction within amino acids of the protein (Rao et al., 2020; Rives et al., 2021). In this work, we select 650-million-parameter (33 layers, 20 heads) and 3-billion-parameter (36 layers, 40 heads) ESM-2 models as baselines and seek to enhance their representation power by applying topological data analysis to their attention maps.

### 3.1 METHOD RES-MST PIPELINE

The overview of our method is presented in Figure 4. Each protein sequence is passed through a protein language model (pLM) to obtain $L \times H$ attention maps, where $L$ is the number of layers in the pLM and $H$ is the number of attention heads per layer.

The attention matrix $A$ of the protein language model denotes the mutual relation of the tokens (amino-acids): the higher the attention is, the stronger is the relation. Each attention matrix is converted to the quasi-distances matrix $W$ computed from attentions: $w_{i,j} = 1 - max(a_{i,j}, a_{j,i})$.

Classical persistent homology extracts multi-scale topological features of the whole graph. However, we are interested in a per-residue predictions, that is, predictions for each node. As we mentioned before, the $H_0$ persistent homology coincides with a MST and can be calculated by a scalable Kruskal's algorithm. Each quasi-distances matrix $W$ is converted into a weighted graph and the set of features is derived from an MST of the weighted graph $\mathcal{G}$. Each node $i$ has a set of incident edges in a MST. Thus, for each node, we calculate *per-RESidue MST* statistics of the incident edges: *min*, *max*, *sum*, *mean* of the weights from incident edges and a *count* of incident edges, denoted as MST features. Besides the features extracted from the MST, we also extract additional features directly from the attention map for each token: self-attention and the sum of absolute values in the $i$-th

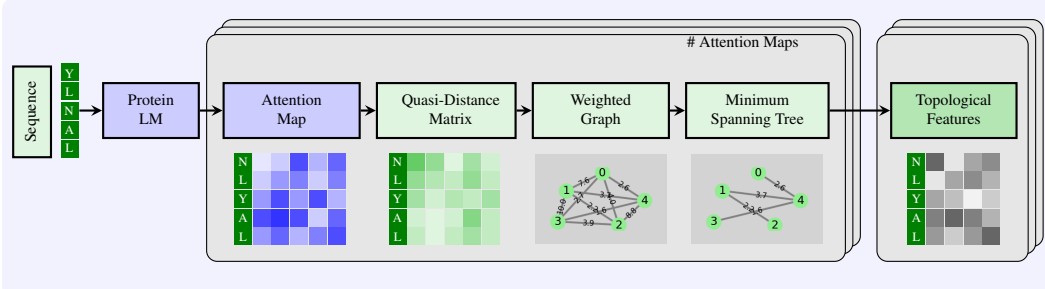

Figure 4: **RES-MST pipeline.** Each protein sequence is processed through a protein language model (pLM) to generate $L \times H$ attention maps, where $L$ is the number of layers in the pLM and $H$ is the number of attention heads per layer. The attention matrices are processed in two ways: either individually for each attention head ($L \times H$ matrices) or averaged across all heads within a layer (producing $L$ matrices). This results in $L \times H$ quasi-distance matrices ($W$) for the RES-MST (all) method or $L$ quasi-distance matrices ($W$) for the RES-MST (avg) method. Each quasi-distance matrix $W$ is subsequently converted into a weighted graph, where the edge weights represent the quasi-distances between amino acids residues. From this weighted graph, a minimum spanning tree (MST) is extracted, *capturing the most significant connections between residues*. Topological features are extracted from the MST for each residue. These features, aggregated across all attention maps, form a comprehensive topological feature set for each residue. Finally, these feature sets are fed into standard machine learning classifiers to predict residue-specific properties.

row and $j$-th column, denoted as non-MST features. These non-MST features are then concatenated with the MST-derived features to form a comprehensive feature vector for each token. The ablation studies between MST and non-MST features can be found in App. section B.1.

We applied topological feature generation to the ESM-2 models under two distinct scenarios:

1. **Individual Attention Matrices**: each of the $L \times H$ attention matrix is processed independently for all distinct attention heads across each layer resulting in $n$ features $\times L \times H$ topological feature set. We refer to this method as RES-MST (all).

2. **Averaged Attention Matrices:** the attention matrices are averaged across all heads within each layer, producing $L$ averaged matrices and resulting in $n$ features $\times L$ topological feature set. We refer to this method as RES-MST (avg).

Topological features derived from the attention maps are aggregated across all/avg attention maps, forming a comprehensive topological feature set for each residue. These feature sets were subsequently fed into standard machine learning classifiers to predict residue-specific properties.

## 3.2 MST NODE DEGREE CORRESPONDS TO A RESIDUE CONSERVATION LEVEL

We begin by studying $H_0$ persistence barcodes. The algorithm for building an $H_0$ persistence barcode is essentially an algorithm for finding a Minimum Spanning Tree (MST) of a weighted graph. Each interval in the $H_0$ barcode corresponds to an edge in MST. To simplify the interpretation of MST, attention matrices of heads in a single layer were averaged.

First, we focus on representative proteins from the conservation test set. Conservative amino acid residues often play significant functional or structural roles in protein folding and activity. To explore the relationship between features obtained from MSTs constructed for different layers of pLM with conservation features, we analyzed these trees alongside the 3D structures of the corresponding proteins, predicted using AlphaFold (Jumper et al., 2021; Varadi et al., 2022). As shown in the analyzed protein set (Figure 5 and Figure 11), across different layers, nodes with the highest degree tend to correspond to residues with high conservation levels. Notably, different conserved residues and graph connectivity patterns were observed across the layers of pLM, highlighting the structural

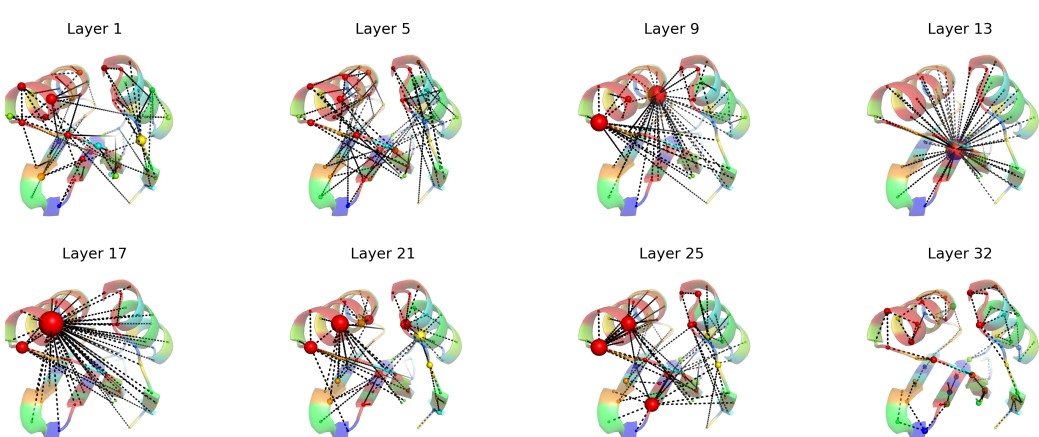

Figure 5: ATX1 metallochaperone protein (Uniprot ID P38636) from the conservation dataset. Minimum spanning trees calculated for the different layers of the pLM transformer are aligned with the 3D structure of the protein. The 3D structures were extracted from the AlphaFold database (Jumper et al., 2021; Varadi et al., 2022). The edges of the graph are represented by dashed lines, while the nodes are depicted as spheres. The radius of the spheres correspond to the logarithm-scaled degree of nodes in the graph. The protein is colored based on conservation, with blue indicating non-conserved residues and red indicating conserved residues.

variability captured at each level. Quantitative analysis revealed a maximum correlation of 0.31 between layers, as shown in Figure 7.

### 3.3 TOPOLOGICAL PATTERNS OF MSTs ENCODE PROTEINS STRUCTURALLY SIGNIFICANT INFORMATION

Next, we study topological patterns of MSTs and how they are mapped onto amino-acid sequences. Figure 6, Figure 8 and Figure 9 present a quantitative analysis. Figure 6 shows a mean maximum degree of a node in MSTs across layers and it has a peak in middle layers and very low values in initial and last layers. Figure 8 and Figure 9 show a mean distance between nodes incident to edges of MTSs: a distance between tokens in a sequence and amino-acids in Euclidean space respectively. These distances are low in last layers. Based on the quantitative analysis, we draw a conclusion on MSTs mapping to protein structure. MSTs corresponding to the **initial layers** are characterized by a "chaotic" connectivity, featuring a greater Euclidean distances between nodes and moderate maximum node degrees. Clearly, in **middle layers** MSTs have a "star" pattern, with one node connecting to all the others. In contrast, the MST constructed for the model's **last layer** exhibits an almost linear configuration, where each node is connected to its immediate neighbors in a sequence.

Based on this analysis, we have concluded that the features extracted through topological analysis of MSTs, corresponding to different layers of the pLM, encode structurally significant information that can be used for addressing specific downstream tasks.

## 4 DATASETS

One way to establish the value of pLMs is to use the vector representations they learned, referred to as the embeddings, as input to subsequent supervised prediction tasks. We evaluate our topological pipeline using open-source datasets on several per-residue tasks: per-residue binding and per-residue conservation. For all prediction tasks in binary or multi-class classification we trained Py-boost classifiers (Iosipoi and Vakhrushev, 2022).

**Conservation.** Residue conservation refers to the phenomenon where specific amino acids in a protein sequence remain unchanged across different species or evolutionary timeframes. Conserva-

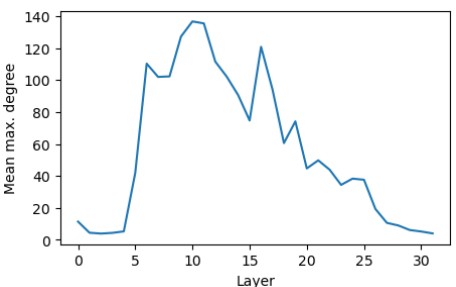

Figure 6: A mean maximum degree of a node in MST.

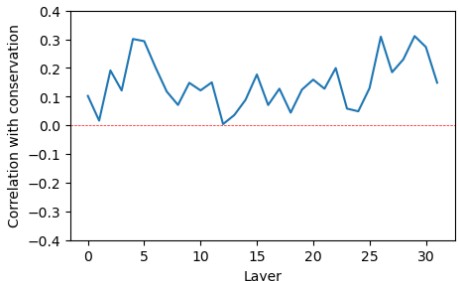

Figure 7: A correlation between a maximum node degree and a conservation value.

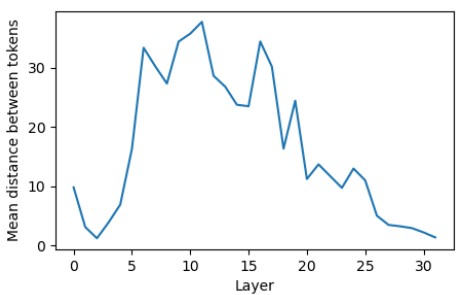

Figure 8: A mean distance between tokens corresponding to incident nodes of edges in MST.

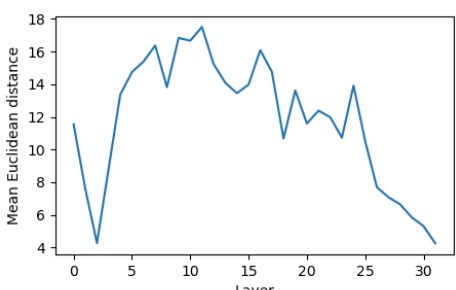

Figure 9: A mean Euclidean distance between amino-acids corresponding to incident nodes of edges in MST.

tion indicates that these residues are critical for the protein's structure and function. Following the previous work (Marquet et al., 2022), we considered ConSurf10k dataset of a well-curated collection of protein sequences annotated with conservation scores. The conservation scores ranged from 1 (most variable) to 9 (most conserved). For the evaluation we followed (Marquet et al., 2022) split of 10,507 proteins into training (9,392 sequences), cross-training/validation (555), and test (519) sets.

**Binding.** Following the work (Yuan et al., 2024), we considered the benchmark of the binding site prediction task across different types of molecules to fairly evaluate and compare the accurate identification of protein binding residues. The benchmark datasets are constructed from BioLiP (Zhang et al., 2024), a database of biologically relevant protein-ligand complexes primarily from the Protein Data Bank (Berman et al., 2000). The benchmark includes annotated sequences for predicting binding residues of the following types of interactions: protein-DNA (DNA), protein-RNA (RNA), protein-peptides (PEP), protein-protein (PRO), protein-ATP (ATP), protein-heme (HEM), and protein-metal ions, such as zinc (Zn2+), calcium (Ca2+), magnesium (Mg2+), and manganese (Mn2+). Combining all these 10 tasks datasets consists of in a total of 8441 training sequences (661 - DNA, 689 - RNA, 1251 - PEP, 335 - PRO, 347 - ATP, 176 - HEM, 1646 - ZN, 1554 - CA, 1729 MG, and 547 MN) and 1838 test sequences (146 - DNA, 346 - RNA, 235 - PEP, 375 - PRO, 79 - ATP, 48 - HEM, 211 - ZN, 183 - CA, 235 MG, and 57 MN).

These tasks of predicting a wide range of protein-ligand interactions are crucial for drug design and understanding protein function, which are key to identifying therapeutic targets and elucidating disease mechanisms. Protein-protein (PRO) and protein-peptide (PEP) interactions are particularly important both for understanding fundamental cellular processes and for drug development, as they play key roles in signaling pathways and structural assembly. ATP and HEM are important metabolites involved in energy transfer and oxygen transport, respectively, making their accurate prediction crucial for understanding metabolic pathways and redox reactions. Predicting DNA and RNA binding sites is essential for understanding gene regulation, transcription, and translation processes, as many proteins involved in these pathways play critical roles in cellular function and disease. Further-

more, metal ions act as major regulators in various biochemical processes, often serving as cofactors for enzymatic reactions, and their binding sites provide insights into protein stability and function in processes like signal transduction and catalysis.

## 5 EXPERIMENTS

We employed the Individual Attention Matrices scenario, RES-MST (all), with the 650-million-parameter ESM-2 model. However, this approach was not applied to the 3-billion-parameter ESM-2 model due to the significant number of topological features it generates, which would lead to increased computational complexity for downstream tasks. In contrast, the Averaged Attention Matrices scenario, RES-MST (avg), was applied to both the 650-million-parameter and 3-billion-parameter ESM-2 models . These feature sets were subsequently fed into Pyboost (Iosipoi and Vakhrushev, 2022) classifier to predict residue-specific properties.

To distinguish the contribution of the topological data analysis approach from leveraging attention patterns, we conducted experiments utilizing a self-attention map aggregation method, as applied to contact map prediction in (Rao et al., 2020). Since we are interested in per-residue predictions, attention maps were summed over rows , in addition to (Rao et al., 2020). This method is referred to as "Attention Map Aggregation."

**Conservation.** Table 1 outlines the results from the conservation prediction task. Our method demonstrated performance, achieving accuracy in nine-states (Q9) per-residue conservation prediction task that is comparable with the traditional alignment strategies, specifically, ConSeq method (Berezin et al., 2004), which relies on multiple sequence alignments (MSAs) (details provided in Section A). Furthermore, our approach excelled in the two-state (Q2, conserved/not conserved) per-residue conservation prediction task, surpassing ESM-2 in terms of accuracy. (Marquet et al., 2022) presented several approaches, which utilize different protein language models than ESM-2 - the ProtBert (Ahmed et al., 2020) and ProtT5-XL-U50 (Ahmed et al., 2020) embeddings, which we did not included for direct comparisons with ESM-2 based approaches.

Table 1: Per-residue conservation prediction experimental results. **Bold** denotes the best performance, *italic* denotes the runner-up. RES-MST (all) method denotes the attention matrices are processed individually for each attention head ($L \times H$ matrices). RES-MST (avg) method denotes the attention matrices averaged across all heads within a layer ($L$ matrices).

| Model | Parameters | Q2 Accuracy (%) | Q9 Accuracy (%) |
|---|---|---|---|
| Random | | $49.9 \pm 0.4$ | $12.4 \pm 0.2$ |
| ConSeq | | $80.2 \pm 0.4$ | *$33.8 \pm 0.2$* |
| ESM-2 | 650M | $79.5 \pm 0.04$ | $33.2 \pm 0.04$ |
| ESM-2 | 3B | *$81.1 \pm 0.02$* | $33.3 \pm 0.03$ |
| RES-MST (all) | 650M | $78.2 \pm 0.02$ | $31.5 \pm 0.02$ |
| RES-MST (avg) | 650M | $75.1 \pm 0.03$ | $27.7 \pm 0.01$ |
| RES-MST (avg) | 3B | $75.9 \pm 0.03$ | $28.4 \pm 0.07$ |
| RES-MST (all) + ESM-2 | 650M | $81.0 \pm 0.02$ | $33.4 \pm 0.01$ |
| RES-MST (avg) + ESM-2 | 650M | $80.9 \pm 0.01$ | $33.2 \pm 0.07$ |
| RES-MST (avg) + ESM-2 | 3B | $\mathbf{81.5 \pm 0.04}$ | $\mathbf{33.9 \pm 0.03}$ |

**Binding.** Table 2 summarizes the results of the binding prediction task. This task is characterized by a significant imbalance in the dataset. To address the imbalance we performed the Synthetic Minority Oversampling Technique (SMOTE) as proposed by (Chawla et al., 2002). Following prior research, we employed the Area Under the Curve (AUC) as the evaluation metric for a comparative analysis. Our approach demonstrates superior performance over standard ESM-2 model embeddings (650M and 3B parameters). Detailed information on the standard deviations in these experiments is provided in the ablation studies, available in Appendix Table 3, Table 4 and Table 5.

Table 2: Per-residue binding prediction experimental results. **Bold** denotes the best performance, *italic* denotes the runner-up. RES-MST (all) method denotes the attention matrices are processed individually for each attention head ($L \times H$ matrices). RES-MST (avg) method denotes the attention matrices averaged across all heads within a layer ($L$ matrices).

| Model | Param. | DNA | RNA | HEM | ATP | CA | MN | MG | ZN | PEP | PRO |
|---|---|---|---|---|---|---|---|---|---|---|---|
| TargetS | - | - | - | 89.2 | 85.5 | 77.6 | 86.4 | 72.4 | 87.4 | - | - |
| Attention map aggregation | 650M | 57.1 | 63.1 | 56.2 | 63.7 | 62.6 | 67.4 | 63.8 | 67.3 | 63.0 | 61.3 |
| Attention map aggregation | 3B | 56.0 | 62.2 | 53.7 | 64.9 | 61.9 | 66.3 | 62.9 | 66.5 | 62.1 | 60.3 |
| ESM-2 | 650M | 86.5 | 85.3 | 91.6 | 89.8 | 82.9 | 93.4 | 76.8 | 96.7 | 74.6 | 69.9 |
| ESM-2 | 3B | 87.9 | 85.7 | 91.7 | 90.5 | 83.4 | 91.7 | 78.5 | 96.5 | 75.1 | 70.3 |
| RES-MST (all) | 650M | 86.0 | 83.7 | 91.2 | 91.6 | *86.4* | **94.7** | *82.4* | 96.9 | 76.2 | 73.2 |
| RES-MST (avg) | 650M | 77.0 | 76.0 | 86.7 | 87.6 | 81.2 | 92.4 | 79.9 | 94.9 | 70.8 | 68.5 |
| RES-MST (avg) | 3B | 77.4 | 75.3 | 86.2 | 87.4 | 82.0 | 92.9 | 79.8 | 95.5 | 71.5 | 69.0 |
| RES-MST (all) + ESM-2 | 650M | *88.3* | 85.8 | *92.4* | **92.4** | **86.9** | *94.4* | **83.4** | **97.2** | *77.8* | *74.4* |
| RES-MST (avg) + ESM-2 | 650M | 88.3 | *85.9* | 92.1 | 91.4 | 85.5 | 93.6 | 82.2 | *97.2* | 76.8 | 73.9 |
| RES-MST (avg) + ESM-2 | 3B | **89.1** | **86.1** | **92.4** | *91.8* | 85.0 | 93.4 | 81.9 | 97.0 | **78.5** | **74.4** |

Other sequence-based methods, such as PepBind (Zhao et al., 2018), PepNN-Seq (Abdin et al., 2022), PepBCL (Wang et al., 2022b), SVMnuc (Su et al., 2019), (Yu et al., 2013) and LMetalSite Yuan et al. (2022) utilize different protein language models than ESM-2 - the ProtBert (Ahmed et al., 2020) and ProtT5-XL-U50 (Ahmed et al., 2020), which we did not included for direct comparisons with the ESM-2 based approaches.

Current state-of-the-art methods, GPSite (Yuan et al., 2024), GraphSite (Shi et al., 2022), GraphBind (Xia et al., 2021), GeoBind (Li and Liu, 2023), ScanNet (Tubiana et al., 2022), DELIA (Xia et al., 2020) and PepNN-Struct (Abdin et al., 2022) leverages 3D structural data of proteins for its training procedure, thus we did not include them into comparison.

## 6 RELATED WORK

**Topological Data Analysis for Deep Learning.** Topological data analysis (TDA) recently started gaining traction in machine learning and deep learning, having a variety of applications (Balabin et al., 2023; Barannikov et al., 2021a;b; Carrière et al., 2020; Hofer et al., 2017; 2019; Hu et al., 2019; Rieck et al., 2018; Trofimov et al., 2023; Zhao and Wang, 2019). Previous works studied topology of attention maps of foundation models in NLP (Cherniavskii et al., 2022; Kushnareva et al., 2021) and speech processing (Tulchinskii et al., 2022). Topology of attention maps proved to be useful for downstream applications: artificial text detection (Kushnareva et al., 2021), acceptability judgement (Cherniavskii et al., 2022) and speech classification (Tulchinskii et al., 2022). Topological data analysis has been performed on 3D structures of proteins (Dey and Mandal, 2018; Koseki et al., 2023; Swenson et al., 2020).

**Protein language models.** The advent of deep learning has revolutionized the field of computational biology, particularly through the development of pLMs (Ahmed et al., 2020; Heinzinger et al., 2023; Lin et al., 2022). These models, often based on the transformer architecture (Vaswani et al., 2017), have been adeptly repurposed to handle the unique challenges presented by protein sequences. Notable among these is the ESM series, with ESM-2 (Lin et al., 2022) trained on the expansive UniRef database (Suzek et al., 2015). This training involves a masked protein modeling approach, where a percentage of amino acids are intentionally obscured and subsequently predicted to train the model (Devlin et al., 2018). This method, similar to techniques used in natural language processing, necessitates the model's understanding of the intricate relationships between amino acids to successfully predict obscured segments.

**Per residue downstream tasks.** We evaluated our method on several binding site and conservation prediction tasks. Most of other works - PepBind (Zhao et al., 2018), PepNN-Seq (Abdin et al., 2022), and PepBCL (Wang et al., 2022b), AttSec (Kim and Kwon, 2023) - used embeddings of protein language models to solve this tasks. AttSec (Kim and Kwon, 2023) also used attention maps as

features transformed by 2D convolution blocks and provided state-of-the-art on secondary structure prediction task.

## 7 LIMITATIONS

While our method offers significant advancements in the application of topological data analysis to protein language models, there are several limitations to consider. **Monomer-Only Focus.** Most of protein language models are trained exclusively on single protein sequences (monomers). Consequently, our approach, which relies on the attention maps generated by these models, is limited to single proteins and does not extend to protein complexes. This restriction may omit the intricate interactions and functionalities that arise in multi-protein assemblies. **Computational Efficiency.** Our method, which integrates topological features with protein language model (pLM) attention maps, is inherently less computationally efficient compared to using pLM embeddings alone (see Appendix C). This trade-off between computational cost and the enhanced representation power provided by topological features is a factor to consider for practical applications.

## 8 CONCLUSION

In this paper, we introduced a novel approach for efficient and accurate per-residue prediction tasks. Our method leverages topological data analysis of attention maps of the transformer-based protein language models. The method outperformed several sequence-based state-of-the-art results across diverse per residue benchmarks and prediction tasks in terms of accuracy. Moreover, our method revealed topological structures in attention maps aligned with biological motifs. This opens up the possibility of expanding this approach to other domains. There are several avenues for future research, such as designing further topologically-aware end-to-end training techniques and investigating the use of more detailed atom-level interactions in a topological manner.

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

# A  TERMS AND DEFINITIONS

**Sequence profiles.** Position Specific Scoring Matrices (PSSMs), commonly referred to as sequence-profiles, are constructed from probabilities assigned to each of the 20 amino acids at every position of an input sequence. These matrices are especially valuable for identifying conservation across sequences: a high value at a position signifies strong conservation, indicative of critical functional or structural roles, whereas a low value suggests weak conservation. PSSMs are derived through Multiple Sequence Alignment (MSA), a widely employed technique that aligns sequences to explore their evolutionary relationships and to understand the structural and functional constraints within protein families. This method is instrumental in enhancing the accuracy of various prediction pipelines in bioinformatics.

**"Star", "linear" and "chaotic" graphs.** "Star graph" - a tree having $k$ nodes, one of them is internal and it is connected to $k-1$ leaves. "Linear graph" - a graph whose vertices can be listed in the order $v_1, v_2, ..., v_k$ such that the edges are $(v_i, v_{i+1})$ where $i = 1, 2, ..., k-1$. "Chaotic" graph is neither "star" or "linear".

# B  ABLATION STUDIES

This section presents multiple ablations we carried out while developing our approach. First, we tested our method solely on per MST features and non-MST features (App. B.1). Second, we constructed a RES-LT method (App. B.2) adaapted for the per residue classification based on $H_0$ and $H_1$ barcodes and tested it under several prediction tasks: conservation (App. B.2.1), binding (App. B.2.2) and secondary structure (App. B.2.3).

## B.1  RES-MST PERFORMANCE WITHOUT NON MST FEATURES

We performed evaluation solely on mst and non-mst features for the binding prediction tasks. The empirical results are shown in Table 3, Table 4 and Table 5.

### B.1.1  RES-MST PERFORMANCE WITHOUT NON-MST FEATURES ON DNA, RNA, HEM AND ATP

We evaluated the RES-MST method using only MST features, excluding non-MST features, for the DNA, RNA, HEM, and ATP binding prediction tasks. The empirical results, presented in Table 3, demonstrate that the performance of RES-MST without non-MST features still surpasses that of ESM-2.

Table 3: Per-residue binding prediction experimental results. **Bold** denotes the best performance, *italic* denotes the runner-up. RES-MST (all) method denotes the attention matrices are processed individually for each attention head ($L \times H$ matrices). RES-MST (avg) method denotes the attention matrices averaged across all heads within a layer ($L$ matrices).

| Model | Para-meters | **DNA** AUC (%) | **RNA** AUC (%) | **HEM** AUC (%) | **ATP** AUC (%) |
|---|---|---|---|---|---|
| ESM-2 | 650M | $86.5 \pm 0.09$ | $85.3 \pm 0.05$ | $91.6 \pm 0.08$ | $89.8 \pm 0.01$ |
| ESM-2 | 3B | $87.9 \pm 0.04$ | $85.7 \pm 0.05$ | $91.7 \pm 0.02$ | $90.5 \pm 0.01$ |
| RES-MST (all) w/o non-MST | 650M | $85.7 \pm 0.04$ | $83.3 \pm 0.03$ | $90.8 \pm 0.02$ | $91.2 \pm 0.02$ |
| RES-MST (all) | 650M | $86.0 \pm 0.07$ | $83.7 \pm 0.06$ | $91.2 \pm 0.03$ | $91.6 \pm 0.01$ |
| RES-MST (avg) w/o non-MST | 650M | $76.4 \pm 0.09$ | $75.3 \pm 0.05$ | $86.1 \pm 0.03$ | $87.1 \pm 0.05$ |
| RES-MST (avg) | 650M | $77.0 \pm 0.06$ | $76.0 \pm 0.02$ | $86.7 \pm 0.05$ | $87.6 \pm 0.04$ |
| RES-MST (avg) w/o non-MST | 3B | $76.8 \pm 0.08$ | $74.7 \pm 0.06$ | $85.8 \pm 0.05$ | $86.7 \pm 0.03$ |
| RES-MST (avg) | 3B | $77.4 \pm 0.06$ | $75.3 \pm 0.08$ | $86.2 \pm 0.07$ | $87.4 \pm 0.01$ |
| RES-MST (all) w/o non-MST + ESM-2 | 650M | $88.0 \pm 0.05$ | $85.6 \pm 0.04$ | $92.1 \pm 0.05$ | $92.0 \pm 0.02$ |
| RES-MST (all) + ESM-2 | 650M | *$88.3 \pm 0.03$* | $85.8 \pm 0.06$ | *$92.4 \pm 0.06$* | **$92.4 \pm 0.01$** |
| RES-MST (avg) w/o non-MST + ESM-2 | 650M | $88.1 \pm 0.02$ | $85.6 \pm 0.06$ | $91.8 \pm 0.04$ | $91.1 \pm 0.03$ |
| RES-MST (avg) + ESM-2 | 650M | $88.3 \pm 0.01$ | *$85.9 \pm 0.08$* | $92.1 \pm 0.06$ | $91.4 \pm 0.02$ |
| RES-MST (avg) w/o non-MST + ESM-2 | 3B | $88.9 \pm 0.06$ | $85.9 \pm 0.03$ | $92.1 \pm 0.04$ | $91.4 \pm 0.04$ |
| RES-MST (avg) + ESM-2 | 3B | **$89.1 \pm 0.07$** | **$86.1 \pm 0.02$** | **$92.4 \pm 0.05$** | *$91.8 \pm 0.03$* |

### B.1.2 RES-MST PERFORMANCE WITHOUT NON MST FEATURES ON CA, MN, MG AND ZN

We evaluated the RES-MST method using only MST features, excluding non-MST features, for the CA, MN, MG and ZN binding prediction tasks. The empirical results, presented in Table 4, demonstrate that the performance of RES-MST without non-MST features still surpasses that of ESM-2.

Table 4: Per-residue binding prediction experimental results on metal ions CA, MN, MG and ZN. **Bold** denotes the best performance, *italic* denotes the runner-up. RES-MST (all) method denotes the attention matrices are processed individually for each attention head ($L \times H$ matrices). RES-MST (avg) method denotes the attention matrices averaged across all heads within a layer ($L$ matrices).

| Model | Para-meters | $CA^{2+}$ AUC (%) | $MN^{2+}$ AUC (%) | $MG^{2+}$ AUC (%) | $ZN^{2+}$ AUC (%) |
|---|---|---|---|---|---|
| ESM-2 | 650M | $82.9 \pm 0.01$ | $93.4 \pm 0.03$ | $76.8 \pm 0.05$ | $96.7 \pm 0.06$ |
| ESM-2 | 3B | $83.4 \pm 0.03$ | $91.7 \pm 0.05$ | $78.5 \pm 0.02$ | $96.5 \pm 0.01$ |
| RES-MST (all) w/o non-MST | 650M | $86.0 \pm 0.05$ | $94.3 \pm 0.02$ | $81.9 \pm 0.03$ | $96.7 \pm 0.01$ |
| RES-MST (all) | 650M | *$86.4 \pm 0.06$* | **$94.7 \pm 0.01$** | *$82.4 \pm 0.02$* | $96.9 \pm 0.01$ |
| RES-MST (avg) w/o non-MST | 650M | $80.8 \pm 0.04$ | $92.1 \pm 0.02$ | $79.5 \pm 0.03$ | $94.5 \pm 0.02$ |
| RES-MST (avg) | 650M | $81.2 \pm 0.05$ | $92.4 \pm 0.01$ | $79.9 \pm 0.02$ | $94.9 \pm 0.01$ |
| RES-MST (avg) w/o non-MST | 3B | $81.7 \pm 0.03$ | $92.5 \pm 0.02$ | $79.3 \pm 0.02$ | $95.2 \pm 0.03$ |
| RES-MST (avg) | 3B | $82.0 \pm 0.02$ | $92.9 \pm 0.01$ | $79.8 \pm 0.01$ | $95.5 \pm 0.01$ |
| RES-MST (all) w/o non-MST + ESM-2 | 650M | $86.4 \pm 0.04$ | $94.1 \pm 0.02$ | $83.1 \pm 0.02$ | $96.9 \pm 0.02$ |
| RES-MST (all) + ESM-2 | 650M | **$86.9 \pm 0.05$** | *$94.4 \pm 0.01$* | **$83.4 \pm 0.01$** | **$97.2 \pm 0.01$** |
| RES-MST (avg) w/o non-MST + ESM-2 | 650M | $85.2 \pm 0.03$ | $93.5 \pm 0.03$ | $81.7 \pm 0.02$ | $96.8 \pm 0.02$ |
| RES-MST (avg) + ESM-2 | 650M | $85.5 \pm 0.02$ | $93.6 \pm 0.06$ | $82.2 \pm 0.01$ | *$97.2 \pm 0.03$* |
| RES-MST (avg) w/o non-MST + ESM-2 | 3B | $84.8 \pm 0.03$ | $93.1 \pm 0.02$ | $81.5 \pm 0.02$ | $96.7 \pm 0.02$ |
| RES-MST (avg) + ESM-2 | 3B | $85.0 \pm 0.02$ | $93.4 \pm 0.04$ | $81.9 \pm 0.04$ | $97.0 \pm 0.03$ |

### B.1.3 RES-MST PERFORMANCE WITHOUT NON MST FEATURES ON PEPTIDES AND PROTEINS

We evaluated the RES-MST method using only MST features, excluding non-MST features, for the peptides and proteins binding prediction tasks. The empirical results, presented in Table 5, demonstrate that the performance of RES-MST without non-MST features still surpasses that of ESM-2

Table 5: Per-residue binding prediction experimental results on peptides and proteins. **Bold** denotes the best performance, *italic* denotes the runner-up. RES-MST (all) method denotes the attention matrices are processed individually for each attention head ($L \times H$ matrices). RES-MST (avg) method denotes the attention matrices averaged across all heads within a layer ($L$ matrices).

| Model | Para-meters | **PEP** AUC (%) | **PRO** AUC (%) |
|---|---|---|---|
| ESM-2 | 650M | $74.6 \pm 0.01$ | $69.9 \pm 0.04$ |
| ESM-2 | 3B | $75.1 \pm 0.04$ | $70.3 \pm 0.07$ |
| RES-MST (all) w/o non-MST | 650M | $75.9 \pm 0.02$ | $72.9 \pm 0.05$ |
| RES-MST (all) | 650M | $76.2 \pm 0.01$ | $73.2 \pm 0.07$ |
| RES-MST (avg) w/o non-MST | 650M | $70.4 \pm 0.02$ | $68.2 \pm 0.05$ |
| RES-MST (avg) | 650M | $70.8 \pm 0.01$ | $68.5 \pm 0.07$ |
| RES-MST (avg) w/o non-MST | 3B | $71.2 \pm 0.05$ | $68.6 \pm 0.04$ |
| RES-MST (avg) | 3B | $71.5 \pm 0.07$ | $69.0 \pm 0.07$ |
| RES-MST (all) w/o non-MST + ESM-2 | 650M | $77.3 \pm 0.03$ | $74.1 \pm 0.04$ |
| RES-MST (all) + ESM-2 | 650M | *$77.8 \pm 0.02$* | *$74.4 \pm 0.06$* |
| RES-MST (avg) w/o non-MST + ESM-2 | 3B | $77.8 \pm 0.05$ | $73.9 \pm 0.02$ |
| RES-MST (avg) + ESM-2 | 3B | **$78.5 \pm 0.07$** | **$74.4 \pm 0.05$** |

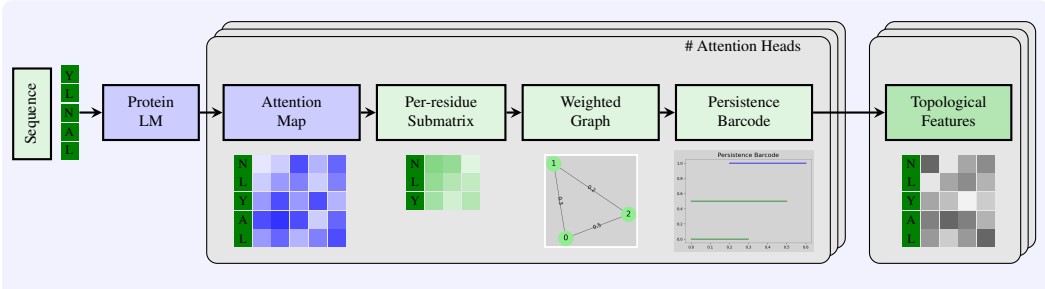

Figure 10: **RES-LT pipeline.** Each protein sequence is passed through a protein language model to obtain attention maps. A submatrix for each residue is extracted by picking neighbors from the quasi-distance matrix $W$. For a specific residue, a localized submatrix is converted into a weighted graph. A persistence barcode is computed from the localized weighted graph. Topological features are extracted from the persistence barcode. Every residue is thus equipped with a topological features set, aggregating information extracted over all attention maps. Finally, residues are classified using a standard machine learning methods on their topological feature set.

## B.2 LOCAL TOPOLOGY PER-RESIDUE FEATURES / METHOD RES-LT

Classical persistent homology extracts multi-scale topological features of the whole graph. However, we are interested in a per-residue predictions, that is, predictions for each node. We hypothesize that for node $i$ only a *local* structure of graph $\mathcal{G}$ matters. For each amino-acid (node) $i$ of the protein sample we identify a neighborhood of $i$ - the subset of close nodes $\{j \mid w_{ij} \leq t\}$ in the graph represented by the quasi-distances matrix $W$ based on the predefined threshold $t$ [2]; together with incident edges, they comprise a fully connected subgraph $\mathcal{G}_i$. Then, for each subgraph $\mathcal{G}_i \subseteq \mathcal{G}$ we calculate its persistence barcode by the `giotto-ph` library (Pérez et al., 2021). We vectorize persistence barcodes by extracting the following features for $H_0$ and $H_1$ barcodes: the number of essential/non-essential bars; the sum, mean, median, max, min and std of bars' lengthes; the entropy of a normalized barcodes; the number of bars having length birth/death smaller than predefined thresholds.

We performed comparison analysis of the RES-LT and RES-MST methods based on ESM-2 650M parameters model on three classification tasks: conservation, binding site and secondary structure prediction tasks.

### B.2.1 CONSERVATION PREDICTION TASK.

Table 6 outlines the results from the conservation prediction task for the dataset defined in section 4.

Table 6: Per-residue conservation prediction experimental results for the RES-MST and RES-LT methods. **Bold** denotes the best performance, *italic* denotes the runner-up. RES-MST (all) method denotes the attention matrices are processed individually for each attention head ($L \times H$ matrices). RES-MST (avg) method denotes the attention matrices averaged across all heads within a layer ($L$ matrices).

| Model | Para-meters | Q2 Accuracy (%) | Q9 Accuracy (%) |
|---|---|---|---|
| ESM-2 | 650M | $79.5 \pm 0.04$ | *$33.2 \pm 0.04$* |
| RES-LT | 650M | $77.7 \pm 0.02$ | $29.5 \pm 0.01$ |
| RES-MST (all) | 650M | $78.2 \pm 0.02$ | $31.5 \pm 0.02$ |
| RES-LT + ESM-2 | 650M | *$80.0 \pm 0.01$* | $33.0 \pm 0.03$ |
| RES-MST (all) + ESM-2 | 650M | **$81.0 \pm 0.02$** | **$33.4 \pm 0.01$** |

---

[2]The threshold was 0.9 in all the experiments.

### B.2.2 BINDING SITE PREDICTION TASK.

**Dataset.** For the comparison analysis of the RES-MST and RES-LT methods we considered two commonly used benchmark datasets (Wang et al., 2022b) to fairly evaluate and compare accurate identification of protein–peptide binding residues.

The first dataset was originally proposed by the study of the structure-based method SPRINT-Str (Taherzadeh et al., 2018). It contains 1,279 protein–peptide complexes with a total of 16,749 binding residues (positive) and 290,943 non-binding residues (negative). From this dataset, they randomly selected 10% complexes as the independent testing set (denoted as TE125), and the remaining as training set (denoted as TR1154). The TE125 set contains 125 proteins with 1,719 binding residues and 29,151 nonbinding residues, while the TR1154 set comprises 1,154 proteins with 15,030 binding residues and 261,792 non-binding residues.

The second dataset is derived from the work (Zhao et al., 2018), which comprises the 1279 protein–peptide complexes with 16,749 peptide-binding residues and 290,943 nonbinding residues. For model training, they randomly chose 640 out of 1,279 complexes with 8,259 binding residues and 149,103 nonbinding residues (denoted as TR640). The remaining complexes with 8,490 binding residues and 141,840 non-binding residues were used as the independent testing set, which is denoted as TE639.

**Results.** Table 7 presents the outcomes of the binding prediction task using sequence-based methods. This task is characterized by a significant imbalance in the dataset. Following prior research, we employed the Area Under the Curve (AUC) as the evaluation metric for a comparative analysis. RES-MST method demonstrates superior performance compared to RES-LT and other sequence-based methods, such as PepBind (Zhao et al., 2018), PepNN-Seq (Abdin et al., 2022), and PepBCL (Wang et al., 2022b). PepNN-Seq leverages a pre-trained contextualized language model, ProtBert (Ahmed et al., 2020), for embedding protein sequences. In a similar vein, PepBCL also utilizes the ProtBert embeddings but adopts a contrastive learning approach to predict protein-peptide binding residues.

Current state-of-the-art methods, PepCNN (Chandra et al., 2023) and PepNN-Struct (Abdin et al., 2022), incorporate additional types of information beyond sequence data alone, thus we did not include them into comparison. Notably, PepCNN combines ProtBert transformer embeddings with sequence-profiles (see definition in Section A) and structural data pertaining to the protein's surface. PepNN-Struct leverages 3D structural data of proteins for its training procedure.

Table 7: Per-residue binding prediction experimental results for the RES-MST and RES-LT methods. **Bold** denotes the best performance, *italic* denotes the runner-up. RES-MST (all) method denotes the attention matrices are processed individually for each attention head ($L \times H$ matrices). RES-MST (avg) method denotes the attention matrices averaged across all heads within a layer ($L$ matrices).

| Model | Parameters | TE125 AUC (%) | TE639 AUC (%) |
|---|---|---|---|
| ESM-2 | 650M | $81.1 \pm 0.02$ | $79.8 \pm 0.03$ |
| RES-LT | 650M | $79.4 \pm 0.04$ | $77.9 \pm 0.01$ |
| RES-MST (all) | 650M | $81.0 \pm 0.06$ | $78.8 \pm 0.04$ |
| RES-LT + ESM-2 | 650M | *$82.0 \pm 0.04$* | *$81.0 \pm 0.01$* |
| RES-MST (all) + ESM-2 | 650M | $\mathbf{83.8 \pm 0.03}$ | $\mathbf{81.9 \pm 0.01}$ |

### B.2.3 SECONDARY STRUCTURE PREDICTION TASK.

**Dataset.** Protein secondary structures can be used as good features for describing the local properties of protein, but also can serve as key features for predicting the complex 3D structures of protein. Thus, it is very important to accurately predict the secondary structure of the protein, which contains a local structural property assigned by the pattern of hydrogen bonds formed between amino acids. The secondary structures are typically assigned by the DSSP (Defne Secondary Structure of Pro-

teins) algorithm (Kabsch and Sander, 1983). The DSSP algorithm checks whether there is hydrogen bond for each amino acid pair by identifying the distance between the elements given the 3D coordinate file of the protein. Then, based on the local patterns of these hydrogen bonds, eight types of secondary structure are assigned to amino acids (DSSP8/Q8): $3_{10}$-helix (G), 4-helix ($\alpha$-helix) (H), 5-helix ($\pi$-helix) (I), hydrogen bonded turn (T), extended strand in parallel and/or anti-parallel $\beta$-sheet conformation (E), residue in isolated $\beta$-bridge (B), bend (S), and coil (C). The aforementioned types can be further grouped into three larger classes (DSSP3/Q3): helix (H), strand (E), and loop (C). While there are several ways to reduce the 8 types to 3 types, we use general reduction: (G/H/I $\rightarrow$ H, E/B $\rightarrow$ E, S/T/C $\rightarrow$ C).

For the comparison analysis of the RES-MST and RES-LT methods we considered the dataset published with NetSurfP-2.0 (Klausen et al., 2019). Following the previous works (Ahmed et al., 2020; Heinzinger et al., 2023; Kim and Kwon, 2023) the training set of the NetSurfP-2.0 contains 10,792 protein sequences both in 3-states (Q3) and 8-states (Q8) DSSP divided into 9,712 training and 1,080 validation data samples. The Py-boost model was trained with NetSurfP-2.0 and then evaluated on NEW364 and CASP12 as well as in (Ahmed et al., 2020; Heinzinger et al., 2023; Kim and Kwon, 2023).

**Results.** Table 8 presents the results for 3-states (Q3) and 8-states (Q8) secondary structure prediction task. We trained our models using the NetSurfP-2.0 dataset (Klausen et al., 2019) and evaluated performance on the NEW364 and CASP12 test sets across different models. Our method was compared with DeepSeqVec (Heinzinger et al., 2019), ProtT5-XL-U50 (Ahmed et al., 2020), ProtT5-XXL-U50, AttSec (Kim and Kwon, 2023), and NetSurfP-2.0. ProtT5-XL-U50 and AttSec use the ProtT5-XL-U50 pretrained language model, which has 3 billion parameters and is based on T5 (Raffel et al., 2020). This model is trained on the UniRef50 (Suzek et al., 2015) dataset via a denoising task from BERT (Devlin et al., 2018), providing 1024-dimension embeddings per token. ProtT5-XXL-U50, also based on T5 and trained on UniRef50, has 11 billion parameters. ProtT5-XXL-U50, ProtT5-XL-U50 and AttSec use the language model embeddings without additional fine-tuning.

RES-MST method, combining topological features with ESM-2 embeddings, demonstrates strong performance in the DSSP Q3 task on both NEW364 and CASP12 test sets, outperforming all other models except AttSec. While AttSec outperforms our method on the NEW364 test set, it is less robust on the CASP12 test set, where our method achieves better results. Also note that AttSec is based on the 3B parameters size model, ProtT5-XL-U50. At the same time, our method utilizes ESM-2 models of 5 times smaller size - 0.65B. The difference with NetsurfP-2.0 for CASP12, DSSP3 is not statistically significant.

In the DSSP Q8 task, our method performs lower than the ProtT5-XL-U50-based methods and NetSurfP-2.0. This is attributed to the larger parameter count of ProtT5-XL-U50 (3 billion vs. 650 million for ESM-2) and NetSurfP-2.0's use of additional evolutionary information. The performance of all aforementioned models was evaluated in terms of accuracy across all test sets.

Table 8: Per-residue secondary structure prediction experimental results. **Bold** denotes the best performance, *italic* denotes the runner-up. RES-MST (all) method denotes the attention matrices are processed individually for each attention head ($L \times H$ matrices). RES-MST (avg) method denotes the attention matrices averaged across all heads within a layer ($L$ matrices).

| Model | Para-meters | Q3 Accuracy (%) | | Q8 Accuracy (%) | |
|---|---|---|---|---|---|
| | | NEW364 | CASP12 | NEW364 | CASP12 |
| ESM-2 | 650M | $84.2 \pm 0.04$ | $81.2 \pm 0.23$ | $73.3 \pm 0.04$ | $69.3 \pm 0.26$ |
| RES-LT | 650M | $82.9 \pm 0.03$ | $79.9 \pm 0.03$ | $70.7 \pm 0.06$ | $66.2 \pm 0.26$ |
| RES-MST (all) | 650M | $82.9 \pm 0.10$ | $80.0 \pm 0.04$ | $71.0 \pm 0.06$ | $66.5 \pm 0.35$ |
| RES-LT + ESM-2 | 650M | *$84.5 \pm 0.01$* | *$81.8 \pm 0.16$* | $73.5 \pm 0.04$ | $69.8 \pm 0.21$ |
| RES-MST (all) + ESM-2 | 650M | **$84.7 \pm 0.01$** | **$81.9 \pm 0.36$** | *$73.7 \pm 0.08$* | *$69.9 \pm 0.13$* |

## C  Computational complexity

For a sequence of size $n$, the attention map has a size of $n \times n$. The derived fully-connected weighted graph has $O(n(n-1)/2)$ edges. The proposed algorithms scale linearly in terms of a network's depth and a number of attention heads.

**RES-LT**. RES-LT method involves a computation of persistence barcodes which is a main source of a complexity. An algorithm is applied to $n$ submatrices of size $n' < n$. The computation is at worst cubic in the number of simplices involved in a simplicial complex. The simplicial complexes are derived from a fully-connected weighted graph of submatrices. In practice, the computation is often quite fast (takes several seconds) for sequences $n < 1024$ since the boundary matrix is typically sparse for real datasets.

**RES-MST**. The complexity of finding a minimum spanning tree with a Kruskal's algorithm is $O(n^2 \, log(n))$. The minimum spanning tree is calculated once for a whole graph and features are generated in $O(n)$ time.

## D  Experiments Compute Resources and Experimental Setting

### D.1  Experiments Compute Resources

For the test set of 1000 proteins and ESM-2 650M parameters model approximate compute resources of the attention maps processing and features generation are following: $\sim 0.5$ hours of attention matrices extraction on the GPU A100, $\sim 16$ hours on Method RES-LT ($\sim 1$ minute per protein) on CPU, $\sim 1.4$ hours on Method RES-MST ($\sim 5$ seconds per protein) on CPU. The computation efficiency for topological features generation is increasing up to 20-40 times (depending on the number of heads per layer in the particular model) when averaging attention maps across all heads within a layer.

### D.2  Experimental Setting

For all downstream tasks we performed classification via the Pyboost classifier with the following hyperparameters: the Sketching strategy is RandomProjectionSketch with using Hessian optimisation, the learning rate is 0.03, the max number of trees is limited to 50k to conservation prediction task and 10k to the binding prediction tasks, max depth is 4, max bin is 256, min data in leaf is 10.

## E  Additional visualizations of minimum spanning trees aligned with 3D structure

In addition to results presented in section 3.2 Figure 5 we present visualizations for three randomly selected proteins from the conservation test set in Figure 11. The analysis suggests that, across various layers of the pLM, nodes with the highest degree frequently correspond to residues with high conservation levels. Furthermore, distinct connectivity patterns were observed in MSTs constructed from different layers, highlighting the structural variability captured at each level.

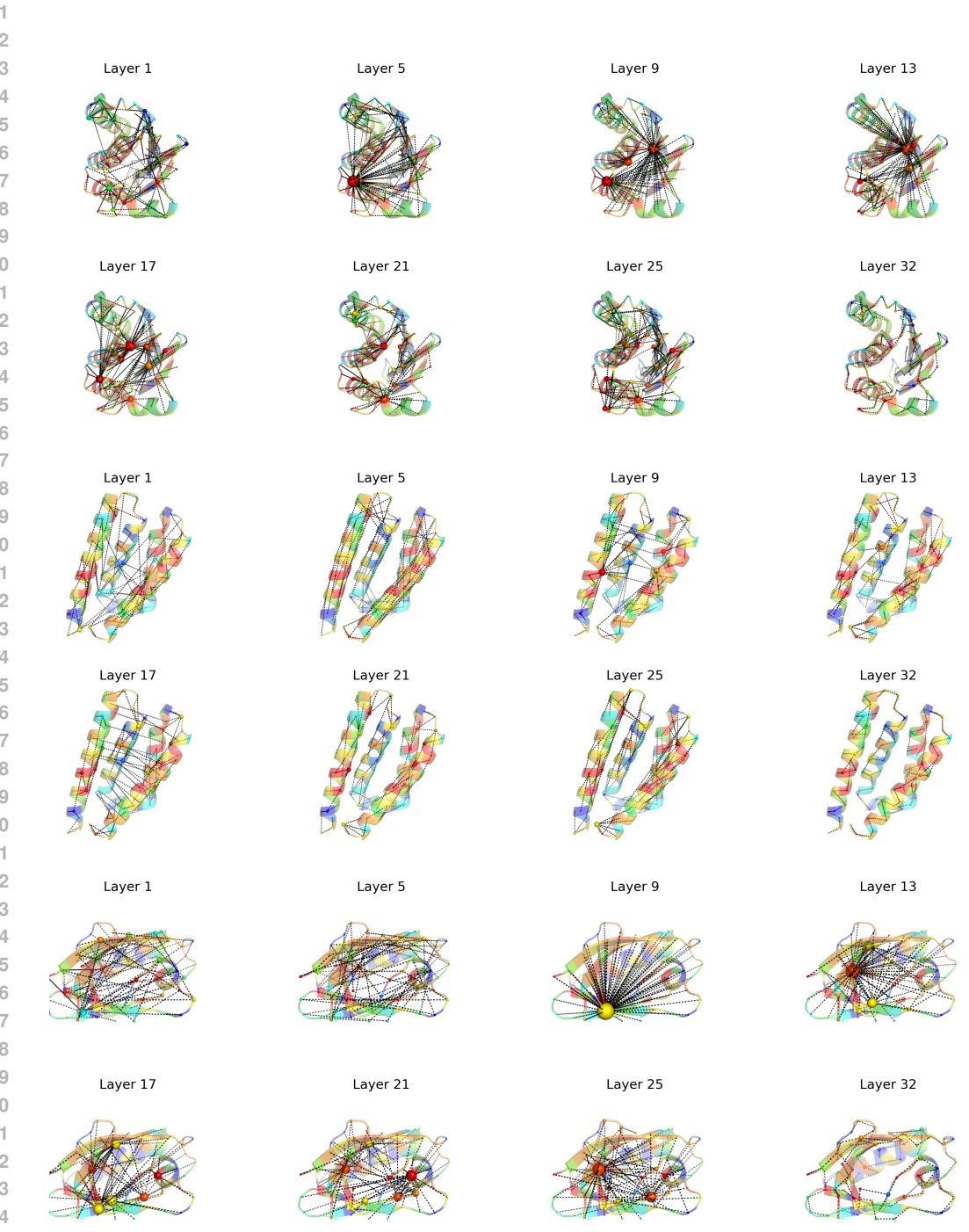

Figure 11: MST node degree corresponds to a residue conservation levels. Minimum spanning trees calculated for the different layers of the pLM transformer are aligned with the 3D structure of the protein. Three randomly selected proteins are shown. The 3D structures were predicted using AlphaFold (Jumper et al., 2021; Varadi et al., 2022). The edges of the graph are represented by dashed lines, while the nodes are depicted as spheres. The radius of the spheres correspond to the logarithm-scaled degree of nodes in the graph. The protein is colored based on conservation, with blue indicating non-conserved residues and red indicating conserved residues.

