# OpenReview forum: "Exploiting Topology of Protein Language Model Attention Maps for Token Classification"
_ICLR.cc/2025/Conference — Submitted to ICLR 2025_

### Official Review · Reviewer_uv5N · 2024-10-28

**Soundness:** 2
**Presentation:** 2
**Contribution:** 2
**Rating:** 5
**Confidence:** 4

**Summary:**

The paper proposes a method that applies the topological information embedded in the attention maps of protein language models (PLMs) to downstream tasks. Specifically, the paper treats the information in the attention maps as a fully connected undirected graph, where each node represents an amino acid. It then extracts a minimum spanning tree (MST) from this graph and further derives effective topological information from the MST to be used in downstream property prediction tasks. In summary, this work represents an effective attempt to mine structure-related topological information from the attention layers in PLMs, offering a new perspective for further analysis and understanding of PLM behavior.

**Strengths:**

1. The research question addressed in this paper is both interesting and significant. Understanding and interpreting the behavior and knowledge learned by PLMs is an essential research direction.
2. The idea of treating the attention map as a fully connected graph and modeling structure-related knowledge by capturing its topological information is innovative and worth investigating.

**Weaknesses:**

1. The paper lacks a more comprehensive discussion of approaches for utilizing the graph topological information in the attention map. While the MST approach is one option, the authors need to provide stronger motivation for choosing this method to capture topological information. For example, why was the MST method chosen? What are its advantages? These questions require further clarification. On this basis, the paper should also offer more comparative and reference experiments, such as evaluating the impact of using different methods to model topological information on performance. The MST inherently tends to capture high-weight edges between each node and its neighbors, akin to capturing information about nodes strongly associated with each node. But what if alternative modeling methods with similar properties were used? For example, one could identify the top k nearest nodes for each node by distance and index, then construct features for downstream tasks. How would this approach differ? I believe this discussion is essential.

2. The paper’s organization needs improvement. The second section introduces substantial background knowledge on topological information, yet this part has little relevance to the content in the following third section. Even if removed, this background section would not impact the understanding of the paper's main content. Furthermore, while defining RES-LT in the appendix, this paper references topological background knowledge from the second section; however, as RES-LT is only used in the appendix, the background knowledge could be moved there as well. In other words, I find the paper's structure to be flawed, with insufficient logical cohesion between different parts.

3. More details regarding the experiments should be provided. For instance, what is the difference between RES-MST (ESM-2 650M all) + ESM-2 (650M) and the RES-MST (ESM-2 650M all) model? I couldn’t find any explanation of this in the paper.

4. The chosen downstream tasks primarily focus on per-residue scale tasks. However, it would be valuable to discuss structure-related tasks on a larger scale (e.g., protein function annotation), as this could reveal whether this MST-based topological modeling approach can capture more global protein property information.

5. A more detailed comparison of the method’s runtime is needed. Compared to traditional full-parameter fine-tuning approaches, your method requires first calculating the MST, then extracting features and training a Pyboost classifier, which incurs significant time costs and may reduce algorithmic efficiency. Therefore, a discussion of the time costs of this approach compared to traditional full-parameter fine-tuning is necessary. However, in Appendix A.5, you did not provide runtime comparisons with baseline models.

**Questions:**

Please refer to the weaknesses section.

---

> ### Author Response · Authors · 2024-11-27
> **Response by Authors. Part 1**
>
> Thank you for the thoughtful review and constructive feedback. We appreciate the recognition of our method’s innovative approach in applying topological analysis to protein language models and its potential to enhance the interpretability and utility of attention maps. Below, we address the key points raised to provide further clarity and reinforce the contributions of our work.
>
> > The paper lacks a more comprehensive discussion of approaches for utilizing the graph topological information in the attention map. While the MST approach is one option, the authors need to provide stronger motivation for choosing this method to capture topological information. For example, why was the MST method chosen? What are its advantages? These questions require further clarification. On this basis, the paper should also offer more comparative and reference experiments, such as evaluating the impact of using different methods to model topological information on performance. The MST inherently tends to capture high-weight edges between each node and its neighbors, akin to capturing information about nodes strongly associated with each node. But what if alternative modeling methods with similar properties were used? For example, one could identify the top k nearest nodes for each node by distance and index, then construct features for downstream tasks. How would this approach differ? I believe this discussion is essential.
>
> Our paper focuses on the application of topological data analysis (TDA) to attention maps derived from protein language models. TDA examines multi-scale structural patterns within data; in our case, this involves analyzing graphs generated from attention maps. Specifically, we study $H_0$​ persistence barcodes, which capture clustering and connectivity patterns at multiple scales. Essentially, these barcodes depict how the clustering patterns of a graph evolve as edges with weights exceeding a varying threshold are removed. We have shown that H_0​ persistence barcodes are essentially equivalent to minimum spanning trees (MSTs); detailed explanations are provided on lines 166–181 in the revised version of the paper. MSTs are efficient to compute, and while many other graph representations exist, they are beyond the scope of our current research.
>
> We have found that certain properties of MSTs, such as maximum node degree, exhibit significant correlation with protein conservation values (Figure 7). This is supported by the consistent performance improvements observed across downstream tasks when integrating MST features with embeddings (Tables 1–2). Although our current approach focuses on $H_0$ barcodes, it can potentially be extended to $H_k, k \ge 1$ barcodes, which capture topological patterns like cycles, 3D voids, etc. A comparative analysis with $H_1$ based method RES-LT , presented in Appendix A.2, demonstrates the advantages of MST-based features, including enhanced predictive performance in binding, conservation, and secondary structure prediction tasks.
>
> > The paper’s organization needs improvement. The second section introduces substantial background knowledge on topological information, yet this part has little relevance to the content in the following third section. Even if removed, this background section would not impact the understanding of the paper's main content. Furthermore, while defining RES-LT in the appendix, this paper references topological background knowledge from the second section; however, as RES-LT is only used in the appendix, the background knowledge could be moved there as well. In other words, I find the paper's structure to be flawed, with insufficient logical cohesion between different parts.
>
> Please refer to a previous answer regarding the relevance of Topological Data Analysis. RES-LT is an alternative method based on topological features. The decision to include the RES-LT performance results in the Appendix is based on the presence of an ablation study section in the supplementary materials, which specifically examines how various features influence performance. Since similar ablation studies for other non-MST feature influences are also presented in the Appendix, placing the RES-LT results there ensures consistency in the paper's structure and presentation.
>
> > More details regarding the experiments should be provided. For instance, what is the difference between RES-MST (ESM-2 650M all) + ESM-2 (650M) and the RES-MST (ESM-2 650M all) model? I couldn’t find any explanation of this in the paper.
>
> Thank you for highlighting this point. We have significantly revised the description of the method (see lines 212-255) and experiments (see lines 384-391) to improve clarity and ensure a more comprehensive explanation. Additionally, we have updated method visualization figure to better align with the revised text.

---

> > ### Comment · Reviewer_uv5N · 2024-11-29
> >
> > Thank you for your response. The revised paper does indeed have a clearer structure. The research topic is interesting, but I still believe that further exploration of runtime and more categories of tasks is necessary. Given these factors, I have increased the score to 5, but I still think the paper requires further improvements.

---

> > > ### Author Response · Authors · 2024-12-04
> > > **Thank you.**
> > >
> > > We appreciate the opportunity to address these questions and are glad we could provide clarifications. Thank you for the insightful review and constructive feedback.

---

> ### Author Response · Authors · 2024-11-27
> **Response by Authors. Part 2**
>
> > The chosen downstream tasks primarily focus on per-residue scale tasks. However, it would be valuable to discuss structure-related tasks on a larger scale (e.g., protein function annotation), as this could reveal whether this MST-based topological modeling approach can capture more global protein property information.
>
> Our primary focus was on per-residue tasks to highlight the ability of RES-MST to capture local structural relationships. Expanding to the  protein function annotation, would be an excellent avenue for future research for our approach.
>
> > A more detailed comparison of the method’s runtime is needed. Compared to traditional full-parameter fine-tuning approaches, your method requires first calculating the MST, then extracting features and training a Pyboost classifier, which incurs significant time costs and may reduce algorithmic efficiency. Therefore, a discussion of the time costs of this approach compared to traditional full-parameter fine-tuning is necessary. However, in Appendix A.5, you did not provide runtime comparisons with baseline models.
>
> Our method’s complexity is detailed in Appendix C, highlighting the linear scaling with respect to depth and attention heads. Compared to full-parameter fine-tuning, our approach offers a computationally efficient alternative by leveraging pre-computed attention maps and using a lightweight classifier for downstream tasks. We will include runtime comparisons with baseline models in future work to further validate efficiency.

---

### Official Review · Reviewer_pPMN · 2024-10-30

**Soundness:** 2
**Presentation:** 2
**Contribution:** 2
**Rating:** 5
**Confidence:** 3

**Summary:**

The authors propose RES-MST, a method that leverages attention maps from protein language models to generate minimum spanning trees (MSTs) and extract various features for per-residue conservation and binding predictions. By evaluating their approach on datasets such as ConSurf10k for conservation and diverse binding site prediction benchmarks, they demonstrate that RES-MST outperforms baseline models, achieving superior accuracy and AUC scores.

**Strengths:**

The paper introduces a non-parametric framework aimed at transforming attention matrices from transformer models into topological features that are customized for token-wise classification. The results presented in the paper demonstrate impressive performance in per-residue conservation and binding predictions. The competitive accuracy and AUC values highlight the effectiveness of the proposed method, particularly in leveraging attention maps from pLMs for generating MSTs.

**Weaknesses:**

Methodology:
- Suitability of attention maps for topology: Attention maps represent learned relationships between sequence tokens (amino acids) based on the model's training objective, which is primarily language-based. These relationships are not necessarily grounded in spatial or physical proximity, which are crucial for understanding protein structure and function. Attention matrices are often dense and noisy, with attention spread across many tokens, which might make topological methods like persistent homology less informative or even misleading when applied naively. The paper would need to convincingly demonstrate that the topology derived from attention maps has a meaningful connection to physical or functional protein properties.
- While the authors analyze MST structures across layers, they don’t provide a clear theoretical or empirical justification for why these specific patterns (chaotic, star, linear) are meaningful in terms of protein functionality or how these differences are expected to relate to biological significance.
- The transformation of attention scores into a quasi-distances matrix is a key step, but the reasoning behind this particular transformation is under-explained. Why the maximum of the bidirectional attention scores is chosen, or how this approach compares with others, isn’t detailed.
- The choice to focus on topological features derived from MSTs lacks sufficient motivation regarding why these features, specifically from MSTs rather than other graph representations.
- The authors do not specify the model used for downstream tasks, nor do they clarify the form and structure of the input to this model. While they detail the process of extracting topological features from attention maps and MSTs, they omit critical information on how these features are subsequently utilized in downstream tasks. Without specifying the model type or its architecture, it’s challenging to assess how effectively the extracted features are integrated or if they are even suited to the task's requirements.

Writing:
- There is no explanation for the choice of the name 'RES-MST.'
- The citation format used in the paper does not adhere to standard conventions. For example, in line 172, the citation 'ESM-2 Lin et al. (2022)' should be formatted as 'ESM-2 (Lin et al., 2022)'. I recommend reviewing and revising the citation style throughout the manuscript.
- There is no interpretation provided for Figures 6, 7, 8, and 9.
- Line 136: there is an incorrect use of the open quotation mark.
- Line 147: "the vertices set" is not grammatically correct.
- Line 150: "The natural issue is a necessity to pick some α." is not grammatically correct.  A grammatically correct version would be: "A natural issue is the necessity of choosing a value for α."
- In the tables, some numbers are in different fonts.

**Questions:**

- There is no explanation for the choice of the name 'RES-MST.' What does 'RES' stand for in 'RES-MST'?
- Could the authors elaborate on the theoretical or empirical rationale behind analyzing MST structures in terms of chaotic, star, and linear patterns? How do these specific patterns relate to protein functionality and biological significance?
- In the paper paper, the author discuss the extraction of topological features from attention maps, but do not specify the model used for downstream tasks. Could you provide more detail about the model type and architecture?

---

> ### Author Response · Authors · 2024-11-27
> **Response by Authors. Part 1**
>
> Thank you for your detailed review and constructive feedback. We have improved the presentation according to suggestions. Below, we address the key points to clarify and strengthen our contribution one by one.
>
> > Suitability of attention maps for topology: Attention maps represent learned relationships between sequence tokens (amino acids) based on the model's training objective, which is primarily language-based. These relationships are not necessarily grounded in spatial or physical proximity, which are crucial for understanding protein structure and function. Attention matrices are often dense and noisy, with attention spread across many tokens, which might make topological methods like persistent homology less informative or even misleading when applied naively. The paper would need to convincingly demonstrate that the topology derived from attention maps has a meaningful connection to physical or functional protein properties.
>
> As mentioned in our paper, several approaches have been proposed for analyzing the attention maps of models trained on protein sequences (Bhattacharya et al., 2021; Vig et al., 2020). According to (Vig et al., 2020) findings, the attention maps generated by the models: highlight amino acid pairs distant in sequence but close in structure, as indicated by correlations with pairwise contacts, highlight binding sites within proteins and capture local secondary structure, revealing patterns corresponding to structural motifs like alpha-helices and beta-sheets. This results suggest that protein language models can infer structural proximity from sequence data alone, recognize functionally important sites essential for protein activity, and detect common structural motifs inherent in protein sequences. This demonstrates the capability of attention maps to uncover intricate structural features solely from sequence information. Based on this analysis we conducted topological data analysis of the attention maps. Our results in Tables 1-2 empirically demonstrate the strong correlation between topological features derived from attention maps and structural/functional properties of proteins. For example, nodes with high degrees in the MST often correspond to highly conserved or biologically critical residues.
>
> Bhattacharya N. et al. Interpreting potts and transformer protein models through the lens of simplified attention //PACIFIC SYMPOSIUM ON BIOCOMPUTING 2022. – 2021.
>
> Vig J. et al. Bertology meets biology: Interpreting attention in protein language models//Ninth International Conference on Learning Representations (ICLR) 2021. – 2021.
>
> > While the authors analyze MST structures across layers, they don’t provide a clear theoretical or empirical justification for why these specific patterns (chaotic, star, linear) are meaningful in terms of protein functionality or how these differences are expected to relate to biological significance.
>
> > Could the authors elaborate on the theoretical or empirical rationale behind analyzing MST structures in terms of chaotic, star, and linear patterns? How do these specific patterns relate to protein functionality and biological significance?
>
> Figures 6-9 illustrate evolving topological patterns across layers, from chaotic to star-like and linear configurations. The patterns align with changes in the functional and structural focus of the transformer layers. For example, the star-like topology highlights the centralization of structural features in intermediate layers, as described in Section 3.3. Figure 6 presents a mean maximum degree of a node in MST and confirms a "star" pattern in middle layers (a high maximum degree) and a "linear" pattern in early and late layers (very low maximum degree). Figure 8 presents a mean distance between tokens corresponding to incident nodes of edges in MST. This value is low in early and late layers, proving a "linear" pattern. See also a visualization of a protein in Figure 5. We are including these explanations in the main text for clarity.
>
> > The transformation of attention scores into a quasi-distances matrix is a key step, but the reasoning behind this particular transformation is under-explained. Why the maximum of the bidirectional attention scores is chosen, or how this approach compares with others, isn’t detailed.
>
> The choice of maximum bidirectional attention ensures symmetry in the quasi-distance matrix, helps to reduce noise and aligns with a symmetric nature of physical residue interactions. We leave an evaluation of alternatives like averaging or independent treating upper- and lower-diagonal parts of attention matrices to further research.

---

> > ### Comment · Reviewer_pPMN · 2024-11-27
> >
> > **Suitability of attention maps for topology:**
> >
> > I appreciate the rationale behind using attention maps to create features for the classifier, as they can convey valuable information, as outlined in your paper. While attention maps are effective at identifying important tokens, much of the information they provide can still be quite noisy. Incorporating features derived from attention maps as complementary inputs alongside representations from foundation models such as ESM-2 could enhance overall performance. However, relying solely on features from attention maps means the model depends entirely on potentially noisy data, which could limit its robustness.
> >
> > To support the reliability of attention maps in this context, stronger evidence would be beneficial. For instance, illustrations of attention scores showing residues that are close in 3D space but distant in sequence receiving high attention could strengthen the argument. While Figure 5 shows something related, it does not clearly highlight such amino acid pairs.
> >
> > I acknowledge the combination of features extracted from attention maps and ESM-2 embeddings in your method. However, the comparable performance of models using only attention-derived features is intriguing and warrants further explanation. If the authors can demonstrate how features derived solely from attention maps can achieve such results—perhaps by showing how attention maps capture information about the "grammar" of protein sequences, as explored in genome data in this work [1]—it could become a key strength of the paper. This finding could provide valuable insights into the quality and relevance of attention maps learned by foundation models like ESM-2. Additional explanations or analyses to clarify why this occurs would greatly enhance the reader's understanding of the approach and its broader implications.
> >
> > Additionally, while Tables 2 and 3 showcase the model’s performance on per-residue conservation and binding prediction, these tasks focus more on residue functions than directly encoding *"structurally significant information about proteins."* As such, they cannot fully demonstrate the ability of attention maps to capture structural properties.
> >
> > **MST patterns (chaotic, star, and linear)**
> >
> > The authors claim that *"the patterns align with changes in the functional and structural focus of the transformer layers"*, but is there a chemically grounded explanation for why these patterns align with the actual functional and structural focus of proteins, if they do? Alternatively, if these patterns do not directly correspond to real protein functions and structures, what might be the underlying reason that learning them leads to improved model performance?
> >
> > **The choice of maximum bidirectional attention**
> >
> > I respectfully disagree with this point. The choice of the transformation method for attention scores is a critical step in this work and requires a robust justification. It is not sufficient to defer this evaluation to future work. If choosing the maximum bidirectional attention is not established as a commonly accepted or proven method that outperforms alternatives, a thorough ablation study with alternative methods should have been conducted within the scope of this work to provide a stronger foundation for the proposed approach. I would appreciate hearing more from the authors on this point to better understand their rationale.
> >
> > [1] DNA language model GROVER learns sequence context in the human genome

---

> > > ### Author Response · Authors · 2024-12-04
> > > **Response to Follow-up Questions**
> > >
> > > > To support the reliability of attention maps in this context, stronger evidence would be beneficial. For instance, illustrations of attention scores showing residues that are close in 3D space but distant in sequence receiving high attention could strengthen the argument.
> > >
> > > > If the authors can demonstrate how features derived solely from attention maps can achieve such results—perhaps by showing how attention maps capture information about the "grammar" of protein sequences, as explored in genome data in this work [1]—it could become a key strength of the paper.
> > >
> > > > The authors claim that "the patterns align with changes in the functional and structural focus of the transformer layers", but is there a chemically grounded explanation for why these patterns align with the actual functional and structural focus of proteins, if they do? Alternatively, if these patterns do not directly correspond to real protein functions and structures, what might be the underlying reason that learning them leads to improved model performance?
> > >
> > > We recognize the importance of these considerations. Relevant analyses will be incorporated into the revised version of the paper.
> > > > Additionally, while Tables 2 and 3 showcase the model’s performance on per-residue conservation and binding prediction, these tasks focus more on residue functions than directly encoding "structurally significant information about proteins." As such, they cannot fully demonstrate the ability of attention maps to capture structural properties.
> > >
> > > To address this, we conducted additional experiments focused on secondary structure prediction, and the findings are included in the revised version of the paper, see Appendix B.2.3.
> > >
> > > > If choosing the maximum bidirectional attention is not established as a commonly accepted or proven method that outperforms alternatives, a thorough ablation study with alternative methods should have been conducted within the scope of this work to provide a stronger foundation for the proposed approach. I would appreciate hearing more from the authors on this point to better understand their rationale.
> > >
> > > We thank you for insightful comments. We performed additional experiments exploring alternative methods. Specifically, instead of maximum bidirectional attention symmetrization, we considered an alternative setting. The elements of an attention matrix are used as weights of a bipartite graph. Then, we calculated the topological features of its minimum spanning tree (MST), a method we refer to as "Bipartite RES-MST" in our analysis. While these experiments are still ongoing, the initial results are promising and suggest that this approach may offer meaningful insights.
> > >
> > > In addition, we implemented an alternative self-attention map aggregation method inspired by the approach used for contact map prediction in Rao et al. (2020). Following their methodology, we applied Average Product Correction (APC) independently on the symmetrized attention maps for each head in the Transformer. Given our focus on per-residue predictions, we extended this approach by summing the attention maps over rows, building on the procedure described in Rao et al. (2020). This method, referred to as "Attention Map Aggregation" performed singificantly worse than RES-MST, see Table 2 in the revised manuscript.
> > > Table 1: Per-residue binding prediction experimental results.
> > > |Model        | Parameters |   DNA   |   RNA  |   HEM   |    ATP  |    CA     |    MN   |     MG   |     ZN   |    PEP  |    PRO  |
> > > |-------------|:-------------:|:--------:|:--------:|:--------:|:--------:|:--------:|:--------:|:--------:|:--------:|:--------:|:--------:|
> > > |Attention map aggregation| 650M|  57.1   |  63.1   |  56.2   |  63.7  | 62.6 |  67.4 | 63.8  | 67.3 | 63.0 | 61.3 |
> > > |Attention map aggregation|   3B   | 56.0 | 62.2  | 53.7  | 64.9 | 61.9  |  66.3 |  62.9  |  66.5  |  62.1 |60.3|
> > > |ESM-2 | 650M |86.5 |85.3 |91.6 |89.8 |82.9 |93.4 |76.8 |96.7 |74.6|69.9|
> > > |ESM-2 | 3B |  87.9  |85.7 |91.7 |90.5 |83.4 |91.7 |78.5 |96.5 |75.1|70.3|
> > > |RES-MST (all) | 650M  |86.0|83.7|91.2 |91.6 |*86.4* | **94.7**| *82.4*| 96.9 | 76.2 |73.2|
> > > |Bipartite RES-MST (all) | 650M  |87.0|84.2|- |- |- | -| -| - | - |-|
> > > |RES-MST (avg) | 650M |77.0 | 76.0  | 86.7  | 87.6  | 81.2  | 92.4 | 79.9 | 94.9  | 70.8  | 68.5  |
> > > |RES-MST (avg) | 3B  | 77.4 |  75.3    | 86.2  | 87.4  | 82.0 | 92.9  | 79.8 | 95.5 | 71.5  | 69.0 |
> > > |RES-MST (all) + ESM-2 | 650M |88.3 |85.8 | *92.4* |**92.4** | **86.9**|*94.4*|**83.4**| **97.2**|*77.8* |*74.4* |
> > > |Bipartite RES-MST (all) + ESM-2 | 650M |**89.2** |**86.2** | - |-| -|-|-| -|- |- |
> > > |RES-MST (avg) + ESM-2 | 650M | 88.3  | 85.9 | 92.1  | 91.4 | 85.5  | 93.6 | 82.2 | *97.2* | 76.8  | 73.9 |
> > > |RES-MST (avg) + ESM-2 | 3B | *89.1*| *86.1*  | **92.4**  | *91.8* | 85.0 | 93.4 |  81.9 | 97.0 |**78.5** |**74.4**|
> > >
> > > [1] Transformer protein language models are unsupervised structure learners (Rao et al., bioRxiv2020/ICLR 2021)

---

> ### Author Response · Authors · 2024-11-27
> **Response by Authors. Part 2**
>
> > The choice to focus on topological features derived from MSTs lacks sufficient motivation regarding why these features, specifically from MSTs rather than other graph representations.
>
> In our paper, we apply topological data analysis to graphs derived from attention maps. In particular, we evaluate a persistent homology of these graphs. The $H_0$ barcode is the simplest and the fastest to compute a type of persistent homology descriptor. In a nutshell, it depicts multi-scale clustering patterns of a graph when removing edges having weight greater than some varying threshold. Essentially, $H_0$ barcodes are equivalent to minimum spanning trees (we provided explanation on that, see lines 166-181 in the revised version of the paper). While many other graph representations exist, they are out of focus of our research.
> Some properties of MST like maximum node degree are significantly correlated with a conservation value of a protein (Figure 7). This is supported by the consistent performance improvements observed across downstream tasks when integrating MST features with embeddings (Tables 1-2).
>
> > The authors do not specify the model used for downstream tasks, nor do they clarify the form and structure of the input to this model. While they detail the process of extracting topological features from attention maps and MSTs, they omit critical information on how these features are subsequently utilized in downstream tasks. Without specifying the model type or its architecture, it’s challenging to assess how effectively the extracted features are integrated or if they are even suited to the task's requirements.
>
> > In the paper paper, the author discuss the extraction of topological features from attention maps, but do not specify the model used for downstream tasks. Could you provide more detail about the model type and architecture?
>
> Thank you for pointing this out, we provided details in section 5. The RES-MST features as well as ESM-2 embeddings are used as input to a PyBoost classifier, a non-parametric model known for its robustness and scalability.
>
> > There is no explanation for the choice of the name 'RES-MST.'
>
> > There is no explanation for the choice of the name 'RES-MST.' What does 'RES' stand for in 'RES-MST'?
>
> Thank you for pointing this out, the choice of the name stands from: for each MST node, we calculate per-RESidue MST statistics. We added this clarification to the revised version of the paper (see lines 212-213 of the revised manuscript).
>
> > The citation format used in the paper does not adhere to standard conventions. For example, in line 172, the citation 'ESM-2 Lin et al. (2022)' should be formatted as 'ESM-2 (Lin et al., 2022)'. I recommend reviewing and revising the citation style throughout the manuscript.
>
> We have adjusted the citation format to standard conventions in the revised manuscript.
>
> > There is no interpretation provided for Figures 6, 7, 8, and 9.
>
> These figures illustrate evolving topological patterns across layers, from chaotic to star-like and linear configurations. The patterns align with changes in the functional and structural focus of the transformer layers. For example, the star-like topology highlights the centralization of structural features in intermediate layers, as described in Section 3.3. Figure 6 presents a mean maximum degree of a node in MST and confirms a "star" pattern in middle layers (a high maximum degree) and a "linear" pattern in early and late layers (very low maximum degree). Figure 8 presents a mean distance between tokens corresponding to incident nodes of edges in MST. This value is low in early and late layers, proving a "linear" pattern. See also a visualization of a protein in Figure 5. We are including these explanations in the main text for clarity.
>
> > - Line 136: there is an incorrect use of the open quotation mark.
> > - Line 147: "the vertices set" is not grammatically correct.
> > - Line 150: "The natural issue is a necessity to pick some α." is not grammatically correct. A grammatically correct version would be: "A natural issue is the necessity of choosing a value for α."
> > - In the tables, some numbers are in different fonts.
>
> Thank you for pointing this out, we have adjusted the open quotation mark, numbers in the tables, changed for “is the vertex set of the graph” and "A natural issue is the necessity of choosing a value for α."

---

### Official Review · Reviewer_WAg8 · 2024-11-01

**Soundness:** 1
**Presentation:** 1
**Contribution:** 2
**Rating:** 3
**Confidence:** 4

**Summary:**

This work introduces a method to extract topological features from protein language model attention maps for improved per-amino-acid classification tasks. The authors present RES-MST, which uses minimum spanning trees derived from attention matrices to capture structurally significant protein information. By combining these topological features with standard embeddings from the PLMs, the method outperforms existing sequence-based approaches on binding site identification and conservation prediction tasks.

**Strengths:**

Novel application of topological data analysis to protein language models: This work bridges two important areas (TDA and protein LMs) in an innovative way, potentially opening up new avenues for analyzing and improving protein language models.

**Weaknesses:**

-	Limited theoretical foundation: The paper lacks a robust theoretical explanation for why this topological approach should outperform alternative methods that leverage attention maps. A stronger motivation for the use of topological data analysis in this context would strengthen the paper's argument.
-	Insufficient ablation studies: The paper would benefit from more comprehensive ablation studies to elucidate the contribution of different components of the method, such as various types of topological features and the impact of different layers.
-	Unclear methodology description: The explanation of the method in Section 3.1 lacks clarity. Specifically: a) The exact features extracted from the MST for each amino acid are not clearly defined. b) The features extracted directly from the attention map are ambiguously described. c) The process of combining the MST-derived and attention map-derived features is not explained. d) The final prediction process using this non-parametric method is not adequately detailed.
-	Ambiguous interpretation of results: The interpretation of Figures 6, 8, and 9 in relation to the described patterns (chaotic, star, linear) in Section 3.3 is not sufficiently clear, making it difficult to follow the authors' reasoning.
-	Choice of evaluation metric for conservation prediction: The authors' decision to treat the conservation prediction task as a classification problem, rather than using regression metrics like Pearson correlation or Spearman's rank correlation, is not well justified.
-	Limited comparison with relevant baselines: The paper lacks comparison with other approaches that use both protein sequence embeddings and their attention maps. This makes it unclear whether the performance improvement stems from the proposed Topological Data Analysis approach or simply from leveraging attention patterns. Additional baselines utilizing both embeddings and attention maps with different methods such as (Rao et al, 2020) is necessary to substantiate the effectiveness of the proposed method.

**Questions:**

- Can you provide more theoretical justification or intuition for why this topological approach should work better than alternative methods that leverage attention maps? How does it capture information that other approaches might miss?
- Could you clarify the feature extraction process in more detail? Specifically: a) What exact features are extracted from the MST for each amino acid? b) What features are extracted directly from the attention map? c) How are these two sets of features combined? d) How is the final prediction made using this non-parametric method?
- The paper describes patterns in the MSTs as "chaotic," "star," and "linear" across different layers. Could you provide a more detailed explanation of how Figures 6, 8, and 9 support these characterizations?
- How does your method compare to other approaches that use both protein sequence embeddings and attention maps? Can you provide additional baselines or comparisons to isolate the contribution of the topological data analysis approach versus simply leveraging attention patterns?

---

> ### Author Response · Authors · 2024-11-27
> **Response by Authors. Part 1**
>
> Thank you for your detailed review and constructive feedback. We have improved the presentation according to suggestions. Below, we address the key points to clarify and strengthen our contribution one by one.
>
> > Limited theoretical foundation: The paper lacks a robust theoretical explanation for why this topological approach should outperform alternative methods that leverage attention maps. A stronger motivation for the use of topological data analysis in this context would strengthen the paper's argument.
>
> > Can you provide more theoretical justification or intuition for why this topological approach should work better than alternative methods that leverage attention maps? How does it capture information that other approaches might miss?
>
> The theoretical foundation of our topological approach is not necessarily that it should consistently outperform alternative methods that leverage attention maps but rather that it complements embeddings to provide additional insights. This success stems from the fact that while ESM-2 embeddings capture rich latent features, they do not explicitly encode the structured, graph-like information present in attention maps. The motivation for our approach lies in the unique ability of topological data analysis (TDA) to capture graph-like, structural relationships within attention maps that are not explicitly represented in embeddings.
> MST-based features, derived from TDA, reflect the inherent topological structure of attention maps. These features have demonstrated strong correlations with biologically significant residues, such as conserved amino acids (Section 3.2), underscoring their biological relevance. By capturing these topological structures, our approach provides an orthogonal perspective to embeddings, offering a richer and more comprehensive representation that enhances downstream predictions.
>
> > Insufficient ablation studies: The paper would benefit from more comprehensive ablation studies to elucidate the contribution of different components of the method, such as various types of topological features
>
> We acknowledge the importance of the comprehensive ablation studies. Recognizing the importance of isolating the contributions of MST-based features from those derived from non-MST features, derived from attention map itself, we have included a detailed ablation study in Appendix B.1. The results reveal that our method using only MST-based features achieves performance comparable to the full RES-MST setup as well significantly outperforming standalone ESM-2 embeddings.
>
> Additionally,  we have conducted comparative analyses of MST-based features (RES-MST) with alternative topological representations, such as local topology features (RES-LT), as presented  in Appendix B.2. These analyses demonstrate the superior performance of RES-MST across tasks like conservation, binding site prediction and secondary structure prediction.
>
> > Ambiguous interpretation of results: The interpretation of Figures 6, 8, and 9 in relation to the described patterns (chaotic, star, linear) in Section 3.3 is not sufficiently clear, making it difficult to follow the authors' reasoning.
>
> > The paper describes patterns in the MSTs as "chaotic," "star," and "linear" across different layers. Could you provide a more detailed explanation of how Figures 6, 8, and 9 support these characterizations?
>
> These figures illustrate evolving topological patterns across layers, from chaotic to star-like and linear configurations. The patterns align with changes in the functional and structural focus of the transformer layers. For example, the star-like topology highlights the centralization of structural features in intermediate layers, as described in Section 3.3. Figure 6 presents a mean maximum degree of a node in MST and confirms a "star" pattern in middle layers (a high maximum degree) and a "linear" pattern in early and late layers (very low maximum degree). Figure 8 presents a mean distance between tokens corresponding to incident nodes of edges in MST. This value is low in early and late layers, proving a "linear" pattern. See also a visualization of a protein in Figure 5. We are including these explanations in the main text for clarity.
>
> > Choice of evaluation metric for conservation prediction: The authors' decision to treat the conservation prediction task as a classification problem, rather than using regression metrics like Pearson correlation or Spearman's rank correlation, is not well justified.
>
> Treating conservation prediction as a classification task aligns with prior work, such as Marquet et al. (2022). However, we acknowledge the potential value of regression metrics (e.g., Pearson or Spearman correlations) and will consider these for future comparisons.
>
> Marquet C. et al. Embeddings from protein language models predict conservation and variant effects //Human genetics. 2022.

---

> ### Author Response · Authors · 2024-11-27
> **Response by Authors. Part 2**
>
> > Unclear methodology description: The explanation of the method in Section 3.1 lacks clarity. Specifically: a) The exact features extracted from the MST for each amino acid are not clearly defined. b) The features extracted directly from the attention map are ambiguously described. c) The process of combining the MST-derived and attention map-derived features is not explained. d) The final prediction process using this non-parametric method is not adequately detailed.
>
> > Could you clarify the feature extraction process in more detail? Specifically: a) What exact features are extracted from the MST for each amino acid? b) What features are extracted directly from the attention map? c) How are these two sets of features combined? d) How is the final prediction made using this non-parametric method?
>
> Thank you for highlighting these areas. We recognize the need for additional clarity and a more detailed methodology description, which has been addressed in the revised version of the paper (see Section 3.1). Additionally, we have updated the visualization of the method pipeline to better illustrate the workflow. Below, we provide a detailed explanation of the feature extraction and prediction process:
>
> a) From the minimum spanning tree (MST) constructed over the attention maps, we extract the following features for each amino acid (represented as a node in the MST): Minimum, maximum, sum, and mean weights of incident edges and node degree (the number of edges connected to the node). These features capture the topological structure and relationships of each amino acid within the attention-based MST.
>
> b) From the attention maps, we extract: self-attention values for each residue and sums of absolute attention values for each row and column of the attention map, reflecting the residue’s importance or centrality in the attention context. These features provide direct insights into how the protein language model encodes relationships between residues.
>
> c) The MST-derived and attention map-derived features are concatenated into a unified feature vector for each amino acid. Feature vectors from multiple attention maps are further concatenated for each amino acid to create a comprehensive representation.
>
> d) The concatenated feature vectors for all amino acids are fed into standard machine learning classifiers to perform per-residue (token-level) classification tasks.
>
> > Limited comparison with relevant baselines: The paper lacks comparison with other approaches that use both protein sequence embeddings and their attention maps. This makes it unclear whether the performance improvement stems from the proposed Topological Data Analysis approach or simply from leveraging attention patterns. Additional baselines utilizing both embeddings and attention maps with different methods such as (Rao et al, 2020) is necessary to substantiate the effectiveness of the proposed method.
>
> > How does your method compare to other approaches that use both protein sequence embeddings and attention maps? Can you provide additional baselines or comparisons to isolate the contribution of the topological data analysis approach versus simply leveraging attention patterns?
>
> Our method is sequence-only, ensuring fair comparisons with ESM-2 embeddings. We excluded structural methods like Rao et al. (2020), which incorporate 3D data. However, we agree that adding baselines leveraging attention maps with sequence embeddings would strengthen our findings and plan to include these in future work.

---

> ### Comment · Reviewer_WAg8 · 2024-11-27
> **Post-Response**
>
> Thank you for the response and clarifications. Unfortunately, however, some key concerns remain:
>
> > Can you provide more theoretical justification or intuition for why this topological approach should work better than alternative methods that leverage attention maps? How does it capture information that other approaches might miss?
>
> > How does your method compare to other approaches that use both protein sequence embeddings and attention maps? Can you provide additional baselines or comparisons to isolate the contribution of the topological data analysis approach versus simply leveraging attention patterns?
>
> While the authors assert the "unique ability of topological data analysis (TDA) to capture graph-like, structural relationships within attention maps that are not explicitly represented in embeddings," this claim is not yet fully substantiated in the paper. Previous works [1-2] have already demonstrated that attention maps of protein models capture structural information. The key question remains: **Is TDA demonstrably more effective at extracting structural information from attention maps?** In the current manuscript, RES-MST is the only method that utilizes both attention maps and embeddings. Thus, it's unclear whether the performance improvement stems from (1) the use of attention maps or (2) the application of TDA. It's possible that alternative methods of leveraging attention maps could yield comparable performance gains.
>
> There appears to be a misinterpretation of Rao et al. (2020). Their use of a limited number of contact map samples (20 sequences) for training the logistic regression component was specific to their focus on contact map prediction. This does not imply that their method always requires structural data. For conservation/binding prediction tasks, it should be feasible to utilize attention maps in a manner similar to Rao et al., without relying on TDA. This approach could serve as a valuable baseline to distinguish the contribution of the topological data analysis approach from simply leveraging attention patterns.
>
> [1] BERTology meets biology: Interpreting attention in protein language models (Vig et al., ICLR 2021)
>
> [2] Transformer protein language models are unsupervised structure learners (Rao et al., bioRxiv2020/ICLR 2021)

---

> > ### Author Response · Authors · 2024-11-30
> > **Response to Follow-up Questions**
> >
> > We appreciate the opportunity to address your concerns and are pleased to provide further insights regarding your follow-up question:
> >
> > > Can you provide more theoretical justification or intuition for why this topological approach should work better than alternative methods that leverage attention maps? How does it capture information that other approaches might miss?
> >
> > Persistent homology (Dey, 2022) is an established tool of Topological Data Analysis which capture shape of data at multiple scales (global and local). It is robust to noise. $H_0$ homology depict multi-scale clustering pattern, $H_1$ homology depicts cycles, $H_2$ depicts voids, etc. In our work, we focus on $H_0$ homology which is fast to compute.
> >
> > > Is TDA demonstrably more effective at extracting structural information from attention maps?
> >
> > To specifically isolate the contribution of the TDA approach compared to leveraging attention patterns directly, we conducted additional experiments using a self-attention map aggregation method, similar to the approach used for contact map prediction in Rao et al. (2020). In our case, since we focus on per-residue predictions, we extended this method by summing attention maps over rows, building on the procedure described in Rao et al. (2020). We refer to this approach as "Attention Map Aggregation" in our analysis:
> >
> > Table 1: Per-residue binding prediction experimental results. **Bold** denotes the best performance, *italic* denotes the runner-up. RES-MST (all) method denotes the attention matrices are processed individually for each attention head ($L \times H$ matrices). RES-MST (avg) method denotes the attention matrices averaged across all heads within a layer ($L$ matrices).
> > |Model        | Parameters |   DNA   |   RNA  |   HEM   |    ATP  |    CA     |    MN   |     MG   |     ZN   |    PEP  |    PRO  |
> > |-------------|:-------------:|:--------:|:--------:|:--------:|:--------:|:--------:|:--------:|:--------:|:--------:|:--------:|:--------:|
> > |Attention map aggregation| 650M|  57.1   |  63.1   |  56.2   |  63.7  | 62.6 |  67.4 | 63.8  | 67.3 | 63.0 | 61.3 |
> > |Attention map aggregation|   3B   | 56.0 | 62.2  | 53.7  | 64.9 | 61.9  |  66.3 |  62.9  |  66.5  |  62.1 |60.3|
> > |ESM-2 | 650M |86.5 |85.3 |91.6 |89.8 |82.9 |93.4 |76.8 |96.7 |74.6|69.9|
> > |ESM-2 | 3B |  87.9  |85.7 |91.7 |90.5 |83.4 |91.7 |78.5 |96.5 |75.1|70.3|
> > |RES-MST (all) | 650M  |86.0|83.7|91.2 |91.6 |*86.4* | **94.7**| *82.4*| 96.9 | 76.2 |73.2|
> > |RES-MST (avg) | 650M |77.0 | 76.0  | 86.7  | 87.6  | 81.2  | 92.4 | 79.9 | 94.9  | 70.8  | 68.5  |
> > |RES-MST (avg) | 3B  | 77.4 |  75.3    | 86.2  | 87.4  | 82.0 | 92.9  | 79.8 | 95.5 | 71.5  | 69.0 |
> > |RES-MST (all) + ESM-2 | 650M |*88.3* |85.8 | *92.4* |**92.4** | **86.9**|*94.4*|**83.4**| **97.2**|*77.8* |*74.4* |
> > |RES-MST (avg) + ESM-2 | 650M | 88.3  | *85.9* | 92.1  | 91.4 | 85.5  | 93.6 | 82.2 | *97.2* | 76.8  | 73.9 |
> > |RES-MST (avg) + ESM-2 | 3B | **89.1** | **86.1**  | **92.4**  | *91.8* | 85.0 | 93.4 |  81.9 | 97.0 |**78.5** |**74.4**|
> >
> > Table 2: Per-residue conservation prediction experimental results. **Bold** denotes the best performance, *italic* denotes the runner-up. RES-MST (all) method denotes the attention matrices are processed individually for each attention head ($L \times H$ matrices). RES-MST (avg) method denotes the attention matrices averaged across all heads within a layer ($L$ matrices).
> > |     Model   | Parameters |  Q2  Accuracy (%)   | Q9 Accuracy (%)  |
> > |-------------|:-------------:|:--------:|:--------:|
> > |Random  | |  49.9 |12.4 |
> > |Attention map aggregation| 650M|56.3 | 15.4 |
> > |Attention map aggregation|3B|58.2 | 16.9 |
> > |ESM-2   | 650M| 79.5 |     33.2 |
> > |ESM-2  | 3B| *81.1*|     33.3 |
> > |RES-MST (all)  | 650M |  78.2 |  31.5 |
> > |RES-MST (avg) | 650M |  75.1 |  27.7 |
> > |RES-MST (avg)  | 3B |  75.9 |  28.4 |
> > |RES-MST (all) + ESM-2    | 650M  |  81.0 |33.4 |
> > |RES-MST (avg) + ESM-2   |  650M |  80.9 |33.2 |
> > |RES-MST (avg) + ESM-2  | 3B  |**81.5**|**33.9**|

---

> > > ### Comment · Reviewer_WAg8 · 2024-12-02
> > > **Post-Response**
> > >
> > > Thanks for the follow-up experiments. However,  It's unclear how exactly the authors used attention maps in "summing attention maps over rows. Thus, I cannot say whether this is a valid baseline. Some of the unanswered questions are:
> > > - What do you mean by "summing attention maps over rows."? -In which axis (row or column), is the Softmax computed for the attention maps?
> > > - How do you deal with the multiple attention maps in each layer and those across different layers? For example, Rao et al. trained a logistic regression on top of attention maps. Have you trained a Py-boost classifiers to aggregate information within multiple attention maps directly?

---

> > > > ### Author Response · Authors · 2024-12-02
> > > > **Response to Follow-up Questions**
> > > >
> > > > > What do you mean by "summing attention maps over rows."? -In which axis (row or column),  is the Softmax computed for the attention maps?
> > > >
> > > > Yes, indeed, clarification is needed here. If we were to simply sum the attention scores over the rows for each attention map, the result would be exactly 1 due to the normalization properties of Softmax. However, following the approach outlined by Rao et al. (2020), we applied Average Product Correction (APC). This adjustment was performed independently on the symmetrized attention maps for each head in the Transformer. As a result, summing over the rows of these corrected attention maps for each head does not yield 1.
> > > >
> > > > > How do you deal with the multiple attention maps in each layer and those across different layers? For example, Rao et al. trained a logistic regression on top of attention maps. Have you trained a Py-boost classifiers to aggregate information within multiple attention maps directly?
> > > >
> > > > To address handling multiple attention maps, we extracted per-residue features from each attention map, resulting in $L \times H$ features for each residue. Similar to Rao et al. (2020), we trained a logistic regression model with $L_1$ regularization, adding further hyperparameter tuning for the regularization term to optimize performance.
> > > >
> > > > We also experimented with Py-Boost classifiers as an alternative to logistic regression for aggregating information from multiple attention maps. However, the Py-Boost approach did not yield significant improvements in performance over logistic regression. Consequently, we chose to retain logistic regression as our primary method, maintaining consistency with Rao et al.'s methodology. That said, we plan to include the Py-Boost results as part of our ablation studies in the final paper to provide a comprehensive analysis.

---

### Official Review · Reviewer_XcsX · 2024-11-02

**Soundness:** 2
**Presentation:** 3
**Contribution:** 2
**Rating:** 3
**Confidence:** 4

**Summary:**

This paper investigates the topological features of protein language model attention maps using the lens of persistent homology. The study computes minimum spanning trees (MSTs) from these attention maps to derive per-residue features. The incorporation of topological features enhances the performance of PLMs in prediction tasks such as residue conservation prediction and binding site prediction. Furthermore, the study analyzes variations in the topological features of the MSTs derived from attention maps across different layers of the language model.

**Strengths:**

- The analysis of attention maps using persistent homology offers a commendable theoretical perspective.
- The relationship between protein attention maps, residue conservation, and amino acid distances has been analyzed extensively.

**Weaknesses:**

- While the paper suggests that the MST method could enhance model performance by extracting topological information from attention maps, it lacks empirical evidence to substantiate this claim. Drawing on prior experience, the potential for performance enhancement with attention map integration appears plausible.
- The benchmarks assessed are less widely used (especially for the conservation prediction task), which challenges the demonstration of the new method's practical applicability.
- The baseline comparisons for the binding experiment are limited in diversity and omit the latest methods (e.g., [1,2]), thereby reducing the persuasiveness of the findings.
- There are many typos in the manuscript. e.g., wrong citation format (e.g., "Several unique properties of proteins can be derived from their 3D structure Wang et al. (2022a); Zhang et al. (2022); Kucera et al. (2024); Sun et al. (2024)." -- the references should be included in a parentheses.) and repetitive figures (e.g., Figure 4 and Figure 10).

[1] Qianmu Yuan, Chong Tian, and Yuedong Yang. Genome-scale annotation of protein binding sites via language model and geometric deep learning. eLife, 13:RP93695, 2024.

[2] Pengpai Li and Zhi-Ping Liu. Geobind: segmentation of nucleic acid binding interface on protein surface with geometric deep learning. Nucleic Acids Research, 51(10):e60–e60, 2023.

**Questions:**

- What's the empirical advantage of the MST-based method in comparison to other deep learning-based methods for topological feature extraction in downstream tasks?

---

> ### Author Response · Authors · 2024-11-27
> **Response by Authors. Part 1**
>
> Thank you for your thoughtful review and detailed feedback. We appreciate your recognition of the theoretical contributions and performance improvements demonstrated in our study. We have improved the presentation according to suggestions. Below we address specific concerns one by one.
>
> > While the paper suggests that the MST method could enhance model performance by extracting topological information from attention maps, it lacks empirical evidence to substantiate this claim. Drawing on prior experience, the potential for performance enhancement with attention map integration appears plausible.
>
> As mentioned in our paper, several approaches have been proposed for analyzing the attention maps of models trained on protein sequences (Bhattacharya et al., 2021; Vig et al., 2020). According to (Vig et al., 2020) findings, the attention maps generated by the models: highlight amino acid pairs distant in sequence but close in structure, as indicated by correlations with pairwise contacts, highlight binding sites within proteins and capture local secondary structure, revealing patterns corresponding to structural motifs like alpha-helices and beta-sheets. This results suggest that protein language models can infer structural proximity from sequence data alone, recognize functionally important sites essential for protein activity, and detect common structural motifs inherent in protein sequences. This demonstrates the capability of attention maps to uncover intricate structural features solely from sequence information. Based on this analysis we conducted topological data analysis of the attention maps.
>
> Our method provides a unique and interpretable perspective by leveraging topological data analysis of attention maps, specifically through minimum spanning trees (MSTs), to enrich traditional embeddings. The topological features extracted from attention maps contain independent information not present in ESM-2  embeddings. Notably, our approach outperforms ESM-2 embeddings in several binding prediction tasks (Tables 2), including protein-metal ion interactions, peptide binding, and protein-protein interactions, demonstrating its practical utility. This success arises because, while ESM-2 embeddings capture rich latent features, they do not explicitly encode the structured, graph-like information inherent in attention maps. By distilling this information, our method captures localized structural relationships and highlights residues that are critical for biological functions, making it a valuable addition to the protein analysis toolkit. Across all experiments (Tables 1-2), combining ESM-2 embeddings with the proposed topological features (RES-MST) consistently outperforms using EMS-2 embeddings alone, with performance improvements across a wide range of tasks (10 types of binding and 2 types of conservation prediction), including a notable +4.9% standalone increase for MG binding prediction.
>
> Bhattacharya N. et al. Interpreting potts and transformer protein models through the lens of simplified attention //PACIFIC SYMPOSIUM ON BIOCOMPUTING 2022. – 2021.
>
> Vig J. et al. Bertology meets biology: Interpreting attention in protein language models//Ninth International Conference on Learning Representations (ICLR) 2021. – 2021.
>
> > The benchmarks assessed are less widely used (especially for the conservation prediction task), which challenges the demonstration of the new method's practical applicability.
>
> We acknowledge the importance of incorporating additional, widely used datasets to better demonstrate the generalizability our approach. To address this, we have included experiments on secondary structure prediction tasks using the NEW364 and CASP12 datasets (Q3 and Q8 prediction tasks) for the RES-MST and RES-LT methods, as presented in the ablation studies (see Table 8).
>
> Additionally, if you have a specific dataset in mind, we would greatly value your suggestion for future evaluations.
>
> > The baseline comparisons for the binding experiment are limited in diversity and omit the latest methods (e.g., [1,2]), thereby reducing the persuasiveness of the findings.
>
> As noted in our paper, we excluded the methods Yuan et al. (2024) and Li & Liu (2023) from our comparisons because they leverage 3D structural data during their training process. To ensure a fair comparison, our method relies exclusively on sequence-based protein language models and does not utilize 3D structural data, which inherently provides more detailed information. The aim of our approach is to demonstrate that while ESM-2 embeddings capture rich latent features, they do not explicitly encode the structured, graph-like information present in attention maps. By extracting and distilling this information, our method captures localized structural relationships and highlights residues critical for biological functions, establishing itself as a valuable tool for protein sequence analysis and prediction tasks.

---

> ### Author Response · Authors · 2024-11-27
> **Response by Authors. Part 2**
>
> > There are many typos in the manuscript. e.g., wrong citation format (e.g., "Several unique properties of proteins can be derived from their 3D structure Wang et al. (2022a); Zhang et al. (2022); Kucera et al. (2024); Sun et al. (2024)." -- the references should be included in a parentheses.) and repetitive figures (e.g., Figure 4 and Figure 10).
>
> Thank you for pointing this out, we have corrected the typos and repetitive elements like Figures, and adjusted citation format to standard conventions in the revised manuscript.
>
> > What's the empirical advantage of the MST-based method in comparison to other deep learning-based methods for topological feature extraction in downstream tasks?
>
> To the best of our knowledge, there are currently no deep learning-based methods specifically designed to extract topological features from attention maps for per-token downstream classification tasks, making direct empirical comparisons unavailable. If the reviewer is aware of such a method, we would greatly appreciate a citation for reference.

---

> > ### Comment · Reviewer_XcsX · 2024-11-29
> >
> > Thank you for your response. However, I remain unconvinced regarding the novelty of the method and its performance. For example, regarding the statement about the comparison with baseline methods: "*It's an unfair comparison because they leverage 3D structural data during their training process*", I find this argument unconvincing. Considering that accessing protein structural information is no longer a significant challenge (e.g., with structure prediction models), I see no compelling reason to differentiate between "methods with structure input" and "methods without structure input" for the same task. Similarly, the claim that "*deep learning-based methods specifically designed to extract topological features from attention maps for per-token downstream classification tasks*" appears puzzling. Why must baselines for "per-token downstream classification tasks" specifically involve "deep learning-based methods designed to extract topological features from attention maps"? In my view, any method capable of performing the same task should be a valid baseline.
> >
> > For these reasons, I am inclined to maintain my original score.

---

> ### Author Response · Authors · 2024-12-02
> **Response to Follow-up Questions**
>
> We are glad that we had the opportunity to address your concerns and would also like to offer further insights regarding your follow-up question.
>
> > However, I remain unconvinced regarding the novelty of the method and its performance.
>
> To the best of our knowledge, our approach is novel, as it represents the first application of topological data analysis (TDA) to protein language model attention maps for per-token classification.
>
> > Considering that accessing protein structural information is no longer a significant challenge (e.g., with structure prediction models), I see no compelling reason to differentiate between "methods with structure input" and "methods without structure input" for the same task.
>
> We understand your perspective regarding the use of structural information, especially with recent advancements in structure prediction models. However, our goal in differentiating between methods that use structural input and those that do not is to emphasize the unique capability of TDA to capture graph-like, structural relationships that are not explicitly encoded in protein embeddings. While integrating structural data could indeed enhance the method, we aim to demonstrate the potential of TDA in extracting structural information independently from explicit structural inputs.
>
> Regarding the comparison with methods utilizing structural data, we recognize that the GPSite model by Yuan et al. (2024) is state-of-the-art in the binding site prediction tasks employed in our experiments. Unfortunately, the training code for GPSite is unavailable, which limits our ability to train GPSite directly on our features rather than on ProtTrans embeddings. Nonetheless, we conducted a feasible comparison test by averaging the prediction scores from both our model and GPSite model. The results show an increase in performance.
>
> |Model        | Parameters |   DNA   |   RNA  |   HEM   |    ATP  |    CA     |    MN   |     MG   |     ZN   |    PEP  |    PRO  |
> |-------------|:-------------:|:--------:|:--------:|:--------:|:--------:|:--------:|:--------:|:--------:|:--------:|:--------:|:--------:|
> |GPSite               |                    |  92.06  |  89.94   |  97.15 | 97.48  | 92.21 |  97.35 | 89.14   | 98.06  | 83.64  | 83.61 |
> |GPSite + RES-MST|   650M   | 92.11    | 89.98    | 97.23 | 97.53   | 92.26 | 97.29 |  89.40  |  98.13  |  83.65 | 83.67 |

---

### Official Review · Reviewer_dMUN · 2024-11-03

**Soundness:** 3
**Presentation:** 3
**Contribution:** 2
**Rating:** 6
**Confidence:** 4

**Summary:**

This paper performs a topological data analysis (TDA) of the attention maps produced by
ESM2 protein language models. Inspired by TDA of natural language model attention maps
and TDA of protein structures, this work leverages the apparent relationship between
attention and 3D structure in ESM models. The authors demonstrate that some topological
features are correlated with structural features of proteins and show that adding
topological features improves per-residue performance on a variety of downstream tasks.
I would recommend to accept this paper. It is difficult to understand the precise
contribution of the TDA methods, but the approach is interesting, and the experiments are
thorough.

**Strengths:**

• The method is an interesting way to probe what ESM models are attending to and
how this relates to its knowledge of 3D structure.
• The figures (both diagrams and renderings) are very clear and helpful.
• The analysis of the relationship between TDA features and 3D structure sheds some
light on the utility of the method.
• The results show a consistent benefit from the method and generally provide a fair
comparison to other state of the art sequence methods.

**Weaknesses:**

The analysis of the TDA in section 3.3 feels somewhat incomplete. Is this just based
on the one example from figure 5? Can some of these descriptions such as
“chaotic” vs “star” or “linear” be quantified? What is the significance of each of
these stages?
• (small) Figure 7 would be clearer if the ymin was set to 0
• LMetalSite, another (strong) sequence-based method from Yuan et al (2024) is
missing from the metal-binding table. Also, it may be appropriate to include
ESMFold-derived structural methods, since this is another sequence
“preprocessing” step.
• The provided source code is incomplete. There was substantial use of a package
called bio_tda which was not provided.

**Questions:**

• Figures 6-9 are interesting, but it is not immediately clear what the takeaway is. It
seems to me that figure 6, 8, and 9 can be explained by: “ESM2 attends more to
linear positional encoding in the early and late layers”.
• What are the specific features included in the RES-MST (*) methods? The
performance of these methods is suspiciously good for just the features listed in
section 3.1 - in particular, the models don’t seem to need residue types?
• How expensive is the MST preprocessing compared to structure prediction with
ESMFold or AlphaFold2?

---

> ### Author Response · Authors · 2024-11-27
> **Response by Authors. Part 1**
>
> Thank you for your detailed and thoughtful feedback. We appreciate your recognition of the novelty and thoroughness of our approach, as well as your constructive suggestions to further clarify and strengthen the paper. We will improve the presentation according to suggestions. Below we address specific concerns one by one.
>
> > The analysis of the TDA in section 3.3 feels somewhat incomplete. Is this just based on the one example from figure 5? Can some of these descriptions such as “chaotic” vs “star” or “linear” be quantified? What is the significance of each of these stages?
>
> We agree that quantifying the stages described as “chaotic,” “star,” and “linear” would add rigor to the analysis. Terms "Star" and "linear" refer to common shapes of "star graph" (a tree having $k$ nodes, one of them is internal and it is connected to $k-1$ leaves) and "linear graph" (a graph whose vertices can be listed in the order $v_1$, $v_2$, ..., $v_k$ such that the edges are $(v_i, v_{i+1})$ where $i = 1, 2, ..., k − 1$. "Chaotic" graph is neither "star" or "linear". We are adding these clarifications to the revised version of the paper (lines 768-771). Quantitative evaluation of “star”, “linear” and “chaotic” patterns is the following. Figure 6 presents a mean maximum degree of a node in MST and confirms a "star" pattern in middle layers (a high maximum degree) and a "linear" pattern in early and late layers (very low maximum degree). Figure 8 presents a mean distance between tokens corresponding to incident nodes of edges in MST. This value is low in early and late layers, proving a "linear" pattern.
>
> >  (small) Figure 7 would be clearer if the ymin was set to 0
>
> Thank you for the suggestion regarding Figure 7. We appreciate your attention to clarity and understand the importance of improving visual presentation. While adjusting $y_{\text{min}}$​ to 0 might enhance readability, Figure 7 illustrates correlation values ranging from -1 to 1, making it crucial to display the full range to accurately represent both positive and negative correlations. This ensures that the graph effectively conveys the significance of correlations exceeding 0, which is essential for interpreting the results correctly.
>
> > LMetalSite, another (strong) sequence-based method from Yuan et al (2024) is missing from the metal-binding table. Also, it may be appropriate to include ESMFold-derived structural methods, since this is another sequence “preprocessing” step.
>
> We acknowledge the omission of LMetalSite (Yuan et al., 2024) from the metal-binding table and appreciate your valuable suggestion. LMetalSite is based on the sequence protein language model prot_t5_xl_uniref50. While it achieves superior performance in metal-binding prediction tasks, it leverages a different protein language model than ESM-2, making direct comparisons with our approach less appropriate. The primary aim of our work is to demonstrate the advantages of our method over standalone embeddings derived from the same protein language model (ESM-2 in our experiments) to ensure a fair and controlled evaluation.
> Similarly, ESMFold explicitly incorporates 3D structural data during training, which inherently contains additional information beyond sequence-based models. As a result, it also cannot be included in fair comparisons with our method, which is designed to rely solely on sequence embeddings.
> In the revised version of the paper, we explicitly reference LMetalSite and ESMFold, clarifying their reliance on different training paradigms and data sources, to provide a more comprehensive and transparent discussion for readers.
>
> > The provided source code is incomplete. There was substantial use of a package called bio_tda which was not provided.
>
> You can find the `bio_tda` package in the `res_mst_tda/src` directory in the supplementary material.
>
> >  Figures 6-9 are interesting, but it is not immediately clear what the takeaway is. It seems to me that figure 6, 8, and 9 can be explained by: “ESM2 attends more to linear positional encoding in the early and late layers”
>
> Yes, you are right. Figure 6 presents a mean maximum degree of a node in MST and confirms a "star" pattern in middle layers (a high maximum degree) and a "linear" pattern in early and late layers (very low maximum degree). Figure 8 presents a mean distance between tokens corresponding to incident nodes of edges in MST. This value is low in early and late layers, proving a "linear" pattern. See also a visualization of a protein in Figure 5. We are adding a more detailed discussion to the paper (300-305).

---

> ### Author Response · Authors · 2024-11-27
> **Response by Authors. Part 2**
>
> > What are the specific features included in the RES-MST (*) methods? The performance of these methods is suspiciously good for just the features listed in section 3.1 - in particular, the models don’t seem to need residue types?
>
> The RES-MST methods utilize a combination of features derived from the MST (e.g., edge weight statistics like min, max, sum, mean, and node degree) and attention-based features (e.g., self-attention values and row/column sums). While residue type information is not directly included, the rich structural signals from the MST features derived from attention maps provide sufficient predictive power, as demonstrated in the results.
>
> > How expensive is the MST preprocessing compared to structure prediction with ESMFold or AlphaFold2?
>
> MST preprocessing is computationally efficient, with a complexity of $O(E \log E)$, where E is the number of edges in the graph. This cost is significantly lower than full structure prediction methods like ESMFold or AlphaFold2, making our approach much faster while still leveraging structural insights from attention maps.

---

### Official Review · Reviewer_KYAV · 2024-11-04

**Soundness:** 2
**Presentation:** 2
**Contribution:** 2
**Rating:** 6
**Confidence:** 3

**Summary:**

The authors apply techniques from topological analysis on graphs to the attention maps of protein language models, (particularly ESM-2)

They generate features that can be appended to the ESM-2 embeddings and then used to help in tasks that make classifications / predictions at the amino acid level.

The authors describe how to generate barcode of different persistent homology features by varying a threshold for edge weights, filtering the edges in the graph and recording when various topological features come and go as the threshold is raised. There are simple edge features (H_0), and cyclical features (H_1), and presumably higher level feature s that can be extracted from the barcode.

Next, the authors state that the topological features for H_0 are equivalent to features derived from the Minimum Spanning Tree (MST).

The experiments show that concatenating these features to the pLM embeddings can improve performance on downstream tasks.

**Strengths:**

The discussion about barcodes and topological features is nice.
The motivation seems clear; An output feature that says 'dense clique around here' or 'cycles present' will be useful for some tasks.

The method is non-parametric which is good, the MST does not need a threshold, and there is no need to fine tune ESM-2.

New features are generated from the attention maps of the transformer network.
Since the transformer was trained on language model tasks, the embedding features at the output do not necessarily encode the graph structure contained in the attention maps, only the information necessary for the output token, so it makes sense to try and include more of the information held in the network of attention weights remaining in the transformer.

The new features do improve the results on downstream tasks.

**Weaknesses:**

The main results only use H_0 features, which can be derived from an MST.
The method for H0 boils down to generating the MST and taking basic statistics over the edges to the neighbors for each node.  There is no description about why these statistics are equivalent to H_0 except [212]: "Each interval in a barcode corresponds to an edge in MST". Perhaps a more thorough description in the appendix could be provided?

When some edge weights are the same, MST can give different resulting graphs, since the order of edges is ambiguous.
And since small changes in attention weights could cause radically different MST doesn't this make the resulting features very noisy? In your experience, how widely spread are transformer attention weights? And how is your method robust to this?

For results using both H_0 and H_1 , one has to look to the appendix for the RES-LT results. While they are not better than MST the main body would be clearer if they were included - there is discussion in section 2 about persistent homology and Betti numbers for H_k, and there is talk of cycles and topological features, but cycles only appear in H_1 and only H_0 is used in all the results (in the main text).

RES-MST takes some statistics over edges are taken per node in the MST. [194] In the description for the features derived from the MST (for each node - min,max,sum,mean weights and count incident edges).
Here it is also mentioned that: "We add: self-attention + sum abs values in ith row jth col.".
There should be an ablation study for the effect these (non-MST) features have.
How much performance do the MST features add over these extra features?

Typos::

*** 159: is
*** a: 201 - LxH should be resulting in L accroding to 187
*** 450 -(2020) - paper title missing.

**Questions:**

What is actual size of resulting feature vector (to be added to ESM-2 Embedding) - 8? or 8 x L (when all heads in layer are averaged).

Perhaps this model has advantages over ESM-2 embeddings because it uses features from other layers in the pLM.
The pretraining task for the pLM is for token reconstruction, which might throw away information about connectivity in the last layer.
What about simply taking ESM-2 features from other the layers (eg. middle + last layers) and concatenating them?

---

> ### Author Response · Authors · 2024-11-27
> **Response by Authors. Part 1**
>
> Thank you for your thorough and constructive feedback. We are pleased that you found the novelty and motivation of our approach compelling, as well as its non-parametric nature and the improvements demonstrated on downstream tasks. We will improve the presentation according to suggestions. Below we address specific concerns one by one.
>
> > The main results only use $H_0$ features, which can be derived from an MST. The method for $H_0$ boils down to generating the MST and taking basic statistics over the edges to the neighbors for each node. There is no description about why these statistics are equivalent to $H_0$ except [212]: "Each interval in a barcode corresponds to an edge in MST". Perhaps a more thorough description in the appendix could be provided?
>
> Thank you for pointing this out. While the equivalence of $H_0$ features to MST-derived statistics is briefly mentioned, we agree that a more detailed explanation would enhance clarity. We are expanding the Section 2  in the revised version of the paper to provide a comprehensive description, see lines 165-181. A bar in the $H_0$ barcode corresponds to an edge in MST, because both are constructed by incrementally connecting components in a graph based on ascending edge weights. In $H_0$ persistent homology, an interval represents the lifespan of a connected component, ending when it merges with another, which directly maps to the addition of an edge in the MST that connects two disjoint components. This equivalence arises because both processes prioritize edges by weight to form a single connected structure. See Section 3.5.3 from (Dey, 2022) for more details.
>
> The basic statistics computed over the edges in the MST, such as the minimum, maximum, sum, and mean of edge weights, align with those derived from the $H_0$​ barcode because the intervals in the barcode encode the same edge weights. The length of each interval in the $H_0$​ barcode corresponds to the weight of an edge in the MST. Therefore, summarizing these weights through statistics directly captures the key features of the $H_0$​ barcode, making the two representations equivalent in terms of the structural information they encode.
>  Dey, T. K., & Wang, Y. (2022). Computational topology for data analysis. Cambridge University Press.
>
> > When some edge weights are the same, MST can give different resulting graphs, since the order of edges is ambiguous. And since small changes in attention weights could cause radically different MST doesn't this make the resulting features very noisy? In your experience, how widely spread are transformer attention weights? And how is your method robust to this?
>
> Thank you, it is a very interesting and insightful question. Persistence barcodes, including an $H_0$ barcode are robust to small perturbations of filtration functions (Skraba, 2020). However individual features which correspond to nodes, like node degree in our algorithm, might change abruptly with small changes of attention maps which are the filtration functions in our case. This is an interesting avenue for further research.
>
> Skraba, P., & Turner, K. (2020). Wasserstein stability for persistence diagrams. arXiv preprint arXiv:2006.16824.
>
> > RES-MST takes some statistics over edges are taken per node in the MST. Here it is also mentioned that: "We add: self-attention + sum abs values in ith row jth col.". There should be an ablation study for the effect these (non-MST) features have. How much performance do the MST features add over these extra features?
>
> Thank you for the suggestion. We acknowledge the importance of distinguishing the contributions of MST-based features from additional self-attention-derived features. To address this, we have included an ablation study in the appendix of the revised version (see Tables 3–5), which provides a detailed analysis of their individual impacts. The results clearly demonstrate that our method using only MST-based features achieves performance comparable to the full RES-MST setup, significantly outperforming the standalone ESM-2 embeddings.
>
> > For results using both H_0 and H_1, one has to look to the appendix for the RES-LT results. While they are not better than MST the main body would be clearer if they were included - there is discussion in section 2 about persistent homology and Betti numbers for H_k, and there is talk of cycles and topological features, but cycles only appear in H_1 and only H_0 is used in all the results (in the main text).
>
> The decision to include the RES-LT performance results in the appendix is based on the presence of an ablation study section in the supplementary materials, which specifically examines how various features influence performance. Since similar ablation studies for other non-MST feature influences are also presented in the appendix, placing the RES-LT results there ensures consistency in the paper's structure and presentation.

---

> ### Author Response · Authors · 2024-11-27
> **Response by Authors. Part 2**
>
> > *** 159: is *** a: 201 - LxH should be resulting in L accroding to 187 *** 450 -(2020) - paper title missing.
>
> Thank you for pointing this out. We have corrected these issues in the revised version of our paper. Specifically, we clarified the notation for $L \times H$ by providing a more detailed explanation (lines 245-251). Furthermore, the missing paper title in line 450 has been added in the revised version.
>
> > What is actual size of resulting feature vector (to be added to ESM-2 Embedding) - 8? or 8 x L (when all heads in layer are averaged).
>
> The resulting feature vector has a size of $7 \times L$ when all heads in each layer are averaged. If all heads across all layers are used, the size becomes $7 \times L \times H$. We have provided a more detailed explanation in the revised version (lines 245–251) to clarify this further. Additionally, we updated the visualization of the method pipeline to better illustrate this process.
>
> > Perhaps this model has advantages over ESM-2 embeddings because it uses features from other layers in the pLM. The pretraining task for the pLM is for token reconstruction, which might throw away information about connectivity in the last layer. What about simply taking ESM-2 features from other the layers (eg. middle + last layers) and concatenating them?
>
> While concatenating embeddings from multiple layers could capture additional information, it would result in extremely large feature vector that poses significant computational and memory challenges.  For example, with ESM-2 (650M),  the feature set size would be $1280 \times 33 = 42,240$, making it nearly infeasible for standard machine learning classifiers. In contrast, our approach produces a much more compact and efficient feature vector: $7 \times 20 \times 33 = 4,620$ when considering all attention heads across layers or $7 \times 33 = 231$ when averaging heads within layers. This significant reduction allows efficient downstream processing while retaining critical information.

---

### Official Review · Reviewer_Mubu · 2024-11-04

**Soundness:** 1
**Presentation:** 3
**Contribution:** 2
**Rating:** 5
**Confidence:** 3

**Summary:**

The authors propose to conduct topological analysis on the attention maps produced by protein language models. The analysis shows a relationship between attention strength and physical contact in 3D structures. The authors then propose to use this extracted tree information to augment protein language models in various tasks.

**Strengths:**

1. The manuscript is well-written. I specifically like the illustration in figure 4 and figure 5.
2. The idea is novel and the analysis is convincing. While it is widely accepted that protein language models can directly and indirectly encode structure information, the effort to directly convert this information into an explicit tree is novel, as far as I know.

**Weaknesses:**

1. The major flaw of this paper is about its experimental results. The table 1 shows minimal improvement over original esm-2. Could authors give a brief explanation? Also, the error reported in these tables is astonishingly low. How are these numbers produced? I think such low variance can only be obtained by training linear modules.

**Questions:**

1. Figure 6-9 should be renamed to one figure with sub figures.
2. The line space of contributions listed in introduction might need adjustment.
3. The "all" and "avg" in table1-4 are not explained in tables' captions.

---

> ### Author Response · Authors · 2024-11-27
> **Response by Authors.**
>
> Thank you for your thoughtful and constructive feedback. We are pleased that you found the novelty and interpretability of our approach compelling and appreciated the effectiveness of our visualisations. We will improve the presentation according to suggestions. Below we address specific concerns one by one.
>
> > The major flaw of this paper is about its experimental results. The table 1 shows minimal improvement over original esm-2. Could authors give a brief explanation?
>
> In terms of experimental results, the topological features extracted from attention maps significantly enhance the accuracy of binding site predictions for various molecular interactions, including protein-metal ions (e.g., CA, MN, MG), protein-protein interactions, and peptides, often surpassing standalone ESM-2. Consequently, our method, RES-MST, achieves notable performance improvements when combined with ESM-2 embeddings, as demonstrated in Tables 1-2. This improvement is driven by our method's ability to distill structured, graph-like information from attention maps, which is not explicitly captured by ESM-2 embeddings. By leveraging these localised structural relationships, our approach effectively identifies critical residues essential for biological functions, adding a valuable dimension to protein sequence analysis and prediction tasks.
> In some cases improvement is large: +4.9% for MG binding prediction, +3.5% for CA binding prediction.
>
> > Also, the error reported in these tables is astonishingly low. How are these numbers produced? I think such low variance can only be obtained by training linear modules.
>
> Standard deviations of metrics are estimated from several runs of a PyBoost classifier with distinct seeds. The PyBoost classifier is designed for robustness and efficiency, resulting in low variance in our reported metrics. This is achieved through careful hyperparameter tuning and effective handling of imbalanced datasets using techniques like SMOTE.
>
> > The line space of contributions listed in introduction might need adjustment.
> The "all" and "avg" in table1-4 are not explained in tables' captions.
>
> In response to your suggestions on presentation, we have adjusted the line spacing for contributions in the introduction and provided detailed explanations for "all" and "avg" in the text (lines 246-251) and in the captions of Tables 1-8 in the revised version of the paper.

---

### Official Review · Reviewer_t9DU · 2024-11-04

**Soundness:** 2
**Presentation:** 2
**Contribution:** 2
**Rating:** 3
**Confidence:** 4

**Summary:**

This paper presents an interesting approach to extract topological features from protein language models. More specifically, they compute the minimum spanning tree (MST) from the attention weights of ESM2. To evaluate their method, they train a PyBoost classifier that takes the processed MST features as input and predict conservation and binding residues. They also ensemble their model with ESM to achieve stronger performance.

**Strengths:**

The authors provide an interesting take on the attention weights. By thinking about it as a fully connected graph, the authors present an interesting analysis using minimum spanning trees.

**Weaknesses:**

Evaluation is limited to binding and conservation.

The proposed models, RES-MST (ESM2-650M all) and RES-MST (ESM2-650M avg), perform comparably with ESM2 across the benchmarks. Specifically, ESM achieves stronger performance in 5 of the 12 benchmarks.

**Questions:**

This paper reports an interesting idea on how to convert attention matrices into topological features. The authors provide analysis and visualizations of the minimum spanning tree on different proteins. They look quite interesting. However, it remains unclear to me what the utility of such an approach is.

Since the topological features are extracted solely from ESM2, ESM2 already contains topological features, albeit in a rich latent representation. The similar performance of the proposed approach and ESM2 seems to suggest that one can implicitly decode these topological features from ESM2. Thus, what is the significance of this approach? Is there anything besides being “the first time that topological data analysis has been applied to classification on a per-token basis”? What are some cases in which the proposed topological features capture information that is not easily accessible from ESM2 embeddings alone? In other words, what are some potential advantages of topological approach over the ESM embedding?

To be clear, the tasks of residue conservation and binding are motivated in the introduction. However, the motivation for topological data analysis is not clear, as ESM seems to perform fine.

---

> ### Author Response · Authors · 2024-11-27
> **Response by Authors.**
>
> Thank you for your thoughtful and insightful review recognizing the novelty of our approach. We will improve the presentation according to suggestions. Below we address specific concerns one by one.
> > Evaluation is limited to binding and conservation.
>
> While our evaluation primarily focuses on binding and conservation prediction tasks (10 types of binding and 2 types of conservation), these were selected as representative examples of biologically significant applications to showcase the utility of our approach. Importantly, the method's flexibility makes it suitable for other tasks, such as protein secondary structure prediction. To demonstrate this, we include experiments on secondary structure prediction in the Ablation Studies section, specifically for the RES-MST and RES-LT methods (see Table 8 in the revised version of our paper).
> > The proposed models, RES-MST (ESM2-650M all) and RES-MST (ESM2-650M avg), perform comparably with ESM2 across the benchmarks. Specifically, ESM achieves stronger performance in 5 of the 12 benchmarks.
>  Since the topological features are extracted solely from ESM2, ESM2 already contains topological features, albeit in a rich latent representation. The similar performance of the proposed approach and ESM2 seems to suggest that one can implicitly decode these topological features from ESM2. Thus, what is the significance of this approach? Is there anything besides being “the first time that topological data analysis has been applied to classification on a per-token basis”? What are some cases in which the proposed topological features capture information that is not easily accessible from ESM2 embeddings alone? In other words, what are some potential advantages of topological approach over the ESM embedding?
>
> Our method provides a unique and interpretable perspective by leveraging topological data analysis of attention maps, specifically through minimum spanning trees (MSTs), which enriches traditional embeddings. Notably, our approach outperforms ESM-2 embeddings in several binding prediction tasks, such as identifying protein-metal ion interactions, peptide binding, and protein-protein interactions, demonstrating its practical utility. This success stems from the fact that while ESM-2 embeddings capture rich latent features, they do not explicitly encode the structured, graph-like information present in attention maps. By distilling this information, our method captures localized structural relationships and highlights residues that are critical for biological functions, making it a valuable addition to the toolkit for protein sequence analysis and prediction tasks.
>
> > However, it remains unclear to me what the utility of such an approach is.
>    However, the motivation for topological data analysis is not clear, as ESM seems to perform fine.
>
> Topological features extracted from attention maps contain an independent information which is missing in EMS-2 embeddings. In all of the experiments (Tables 1-2) combining ESM-2 embeddings with the proposed features (RES-MST) is better than using EMS-2 embeddings alone. The utility of our approach is the improved performance in a wide range of tasks (10 types of binding, 2 types of conservation prediction). The improvement is up to +4.9% for MG binding prediction.

---

### Author Response · Authors · 2024-12-04
**Overall Response by Authors**

We thank the reviewers for their detailed and thoughtful reviews. We are pleased to see that our approach has been generally recognized as novel and compelling. We have addressed individual questions, comments, and concerns in separate threads.

Below, we summarize the main revisions to the manuscript for the convenience of the reviewers and the AC:

- The manuscript has been extensively revised to enhance clarity and readability. We provided a more detailed description of the feature extraction process from attention maps and Minimum Spanning Trees (MSTs), as well as a clearer explanation of the workflow for downstream tasks, including feature combination and classifier design. Additionally, the topological patterns across transformer layers—ranging from chaotic to star-like to linear configurations—are now explained with greater precision.
- Comprehensive ablation studies were conducted to isolate the contributions of different features, including those derived from MSTs, non-MST features. These results, presented in Appendix B.1, demonstrate that MST-based features provide significant value (Appendix B.1).
- Beyond per-residue binding and conservation tasks, we included experiments on secondary structure prediction (Q3/Q8 tasks) using the NEW364 and CASP12 datasets. These results (see Appendix B.2.3) highlight the broader applicability of the proposed method.
- To specifically isolate the contribution of the topological data analysis (TDA) approach compared to directly leveraging attention patterns, we conducted additional experiments utilizing a self-attention map aggregation method inspired by the approach used for contact map prediction in Rao et al. (2020).
- Regarding the state-of-the-art model for binding site prediction tasks, which leverages additional structural data, we conducted a feasible comparison test by averaging the prediction scores with our model. This combined approach demonstrated an increase in performance.
- Additionally, we conducted additional experiments of the alternative approach to the symmetrization of attention maps, such as bipartite graph.

We believe that these additional results further strengthen our work, and we sincerely thank the reviewers for their insightful suggestions and constructive feedback.

---

### Meta-Review · Area_Chair_i2iG · 2024-12-22

**Metareview:**

**Summary:**

 Topological Data Analysis (TDA) encompasses topological descriptors (capturing features such as number of components/clusters, number of independent loops) that can augment the capabilities of deep neural architectures. This work proposes to enhance the representations in Transformer-based protein language models (PLMs) with topological features extracted from their attention maps via persistent homology (PH). Specifically, attention weights are treated to get edge weights in a fully connected graph, and a threshold-based filtration is applied to obtain a persistence barcode. While the barcode can be used to extract higher-order information such as that pertaining to cycles, this work focuses only on the zero order features, for each amino acid (i.e., a token in the language model), efficiently computed with a Minimum Spanning Tree (MST) algorithm.  Empirically, the authors show that combining these embeddings with those from an ESM-2 model can improve performance on multiple tasks such as binding site identification and conservation prediction tasks.


**Strengths:**
Reviewers generally acknowledged many strong points of this work, including, (a) clarity of presentation (writing; illustrations; background on topological features) though some also pointed out parts where readability and clarity could be enhanced, (b) interesting and important research problem (clear motivation with a potential to help better understand what PLMs learn; can potentially capture global structural information that PLMs on their own might miss out on), and (c) innovative way to encode attention (bridging TDA and PLMs; no need to finetune PLMs), and (d) empirical benefits on binding site identification and conservation prediction tasks.


**Weaknesses:**
Reviewers also raised several concerns, including, (a) lack of analysis on topological stability (i.e, robustness to small perturbations in the attention weights),  (b) evaluation limited only to per-amino acid scale tasks, i.e.,  binding and conservation tasks (no evidence for structural tasks such as protein function annotation), (c) benchmarks being less common, (d)  lack of comparison with recent methods, as well as prior approaches (such as Rao et al. 2020)  that use both the usual token embeddings and the attention maps, (e) issues with interpretation of some results and choice of metrics, (e) insufficient ablation (e.g,, impact of the number of layers), (f) discussion of alternative approaches than MST for extracting topological information from the attention weights, and (g) lack of detailed comparison on runtime. Some reviewers also pointed out that the empirical improvements over the original ESM-2 were not persuasive,


**`Recommendation:**

The authors addressed many of the above concerns and clarified some aspects during their discussion with the reviewers. However, some key issues remained unresolved.

Reviewer XcsX maintained that since 3D structural information about proteins is accessible these days, the authors’ argument about comparing with methods that use such structural features being unfair is not compeling.  I strongly agree with the reviewer’s assessment that the work loses considerable significance absent comparisons  with other methods that are capable of performing the same tasks,

Reviewer uv5N emphasized that more comprehensive evaluations including on runtime and with additional tasks was necessary. I endorse this feedback.

Finally, reviewer pPMN also underscored the need for stronger evidence such as the attention solely being able to demonstrate in their setting that long-term dependencies in the sequence are localised in the 3D space (e.g., in the context of protein design, such as observation was made in a PLM by Ingraham et al., Generative models for graph-based protein design, NeurIPS (2019)).   This they argued would convincingly demonstrate the ability of the current approach to capture the underlying generative grammar of the protein sequences. They also raised another important point that residue-wise classification should not be restricted to methods like theirs that extracts topological features from attention,  I fully support these concerns.

Without the resolution of these major shortcomings, this work does not meet the bar for acceptance.

**Additional Comments On Reviewer Discussion:**

Please see the Metareview above for all the relevant details.

---

### Decision · Program_Chairs · 2025-01-22

Reject